# LANCE: Low Rank Activation Compression for Efficient On-Device Continual Learning

## Abstract

On-device learning is essential for personalization, privacy, and long-term adaptation in resource-constrained environments. Achieving this requires efficient learning, both fine-tuning existing models and continually acquiring new tasks without catastrophic forgetting. Yet both settings are constrained by high memory cost of storing activations during backpropagation. Existing activation compression methods reduce this cost but relying on repeated low-rank decompositions, introducing computational overhead. Also, such methods have not been explored for continual learning. We propose LANCE (Low-rank Activation Compression), a framework that performs one-shot higher-order Singular Value Decomposition (SVD) to obtain a reusable low-rank subspace for activation projection. This eliminates repeated decompositions, reducing both memory and computation. Moreover, fixed low-rank subspaces further enable on-device continual learning by allocating tasks to orthogonal subspaces without storing large task-specific matrices. Experiments show that LANCE reduces activation storage up to $250\times$ while maintaining accuracy comparable to full backpropagation on CIFAR-10/100, Oxford-IIIT Pets, Flowers102, and CUB-200 datasets. On continual learning benchmarks (Split CIFAR-100, Split MiniImageNet, 5-Datasets), it achieves performance competitive with orthogonal gradient projection methods at a fraction of the memory cost. These results position LANCE as a practical and scalable solution for efficient fine-tuning and continual learning on edge devices.

## 1 Introduction

On-device learning can be a key enabler for personalization, privacy preservation, and rapid adaptation in resource-constrained environments. Unlike cloud-based training, on-device learning allows models to adapt directly on hardware such as smartphones, IoT devices, or embedded systems, ensuring that data remain local and inference remains low-latency (Ren et al., 2024). Crucially, many real-world scenarios require not only efficient fine-tuning but also continual learning, where models incrementally acquire new tasks without catastrophic forgetting (Wang et al., 2024; Kudithipudi et al., 2022; Hadsell et al., 2020; Ke et al., 2021). Combination of efficiency and adaptability is essential for edge devices that must operate over long lifetimes while processing evolving user data streams. Yet, both fine-tuning and continual learning are severely constrained by the large memory footprint of storing intermediate activations during backpropagation. For modern deep neural networks, activation memory often exceeds parameter memory by $5$–$10\times$ (Cai et al., 2020; Lin et al., 2022a), making training infeasible on low-power devices such as microcontrollers with only a few hundred kilobytes of Static Random Access Memory (SRAM) (Ankit et al., 2020; Lin et al., 2022a).

Several approaches have been proposed to address this issue. Checkpointing (Chen et al., 2016) reduces memory usage at the cost of extra recomputation; reversible networks (Gomez et al., 2017) eliminate the need to store activations but require specialized architectures; and quantization or activation pruning (Chen et al., 2021; Barley & Fröning, 2023; Jiang et al., 2022) compress or sparsify activations but may introduce approximation errors or hardware overhead. Alternatively, bio-inspired methods have explored efficient replacements for BP, such as forward-forward methods (Dellaferrera & Kreiman, 2022; Hinton, 2022) and local learning rules (Lillicrap et al., 2014; Nøkland, 2016; Apolinario et al., 2025b; Frenkel et al., 2021; Apolinario et al., 2025c). They usually incur accuracy degradation. More recently, low-rank approaches exploit redundancy in activations using SVD or Tucker decompositions (Nguyen et al., 2024; 2025), but these typically recompute de-

compositions at each training step, adding significant overhead and making them unsuitable for continual learning, where repeated factorization across tasks would require storing large task-specific matrices.

A key insight from orthogonal gradient projection methods in continual learning (Saha et al., 2021; Liang & Li, 2024; Apolinario et al., 2025a) is that neural activations often lie in a stable, low-rank subspace that can be reused across training. This suggests a natural question: *Can we identify a compact activation subspace and reuse it throughout fine-tuning, while simultaneously enabling efficient continual learning?*

With the above in mind, we introduce LANCE (Low-rank Activation Compression), a framework that applies one-time higher-order singular value decomposition (HOSVD) to extract a reusable low-rank activation subspace, as shown in Fig. 1b. Unlike prior iterative approaches (Nguyen et al., 2024; 2025), LANCE computes the decomposition only once at the beginning of training. Subsequent activations are projected onto this fixed subspace throughout training, avoiding repeated decompositions, thereby, significantly reducing both memory usage, computational cost, and hardware overhead. In contrast to system-level methods such as TinyTL (Cai et al., 2020) or the 256KB training engine (Lin et al., 2022a), which freeze layers or restrict updates to fit memory budgets, LANCE enables full backpropagation across all layers by directly compressing stored activations. Importantly, these fixed subspaces also allow assigning new tasks to orthogonal components, inherently supporting on-device continual learning without storing large task-specific matrices as in previous continual learning (CL) methods (Saha et al., 2021; Saha & Roy, 2023; Apolinario et al., 2025a; Lin et al., 2022b;c; Zhao et al., 2023; Shang et al., 2025). By eliminating repeated decompositions while preserving full-model fine-tuning, LANCE provides a practical and scalable solution for memory- and energy-efficient on-device continual learning.

Our main contributions are summarized as follows:

- We propose LANCE, a one-shot activation compression framework that constructs a reusable low-rank subspace via HOSVD, avoiding repeated decompositions.
- We show that LANCE enables both efficient fine-tuning and continual learning on edge devices by projecting activations onto fixed low-rank subspaces, which can be partitioned across tasks to mitigate forgetting.
- We empirically validate LANCE on fine-tuning benchmarks (CIFAR-10/100, Pets, Flowers102, CUB-200 with MCUNet, MobileNetV2, ResNet18/34) and continual learning benchmarks (Split CIFAR-100, Split MiniImageNet, 5-Datasets), demonstrating memory savings of up to $250\times$ while maintaining accuracy within $2\%$ of full backpropagation and achieving competitive performance with full-rank gradient projection methods at a fraction of the memory cost.

## 2 METHODOLOGY

In this section, we outline our methodology. We first describe the main memory bottleneck for on-device learning with full backpropagation (BP), then review higher-order singular value decomposition (HOSVD), and finally present our proposed method, including an overview, step-by-step pseudocode, a complexity analysis, and a convergence analysis.

### 2.1 PRELIMINARIES

Training deep neural networks requires storing intermediate *activations* during the forward pass so that they can be reused for gradient computation in backpropagation, as illustrated in Fig. 1a. Let $f(\mathbf{x}; \boldsymbol{\theta})$ denote a neural network with input $\mathbf{x}$ and parameters $\boldsymbol{\theta}$. During forward propagation, each layer $l$ produces an activation tensor $\mathbf{X}^{(l)}$ that is stored in SRAM for later use in computing the gradient: $\nabla_{\boldsymbol{\theta}^{(l)}} \mathcal{L} = \frac{\partial \mathcal{L}}{\partial \mathbf{X}^{(l)}} \frac{\partial \mathbf{X}^{(l)}}{\partial \boldsymbol{\theta}^{(l)}}$, where $\mathcal{L}$ is the loss function. The cumulative memory footprint of storing $\{\mathbf{X}^{(l)}\}_{l=1}^{L}$ across all $L$ layers often dominates SRAM usage, especially in convolutional networks. For example, on CIFAR-100 with ResNet18, storing activations during training can consume up to $5\times$ more memory than storing weights, quickly exhausting the limited resources of edge devices. A natural way to mitigate this bottleneck is to exploit the *low-rank structure* of activations.

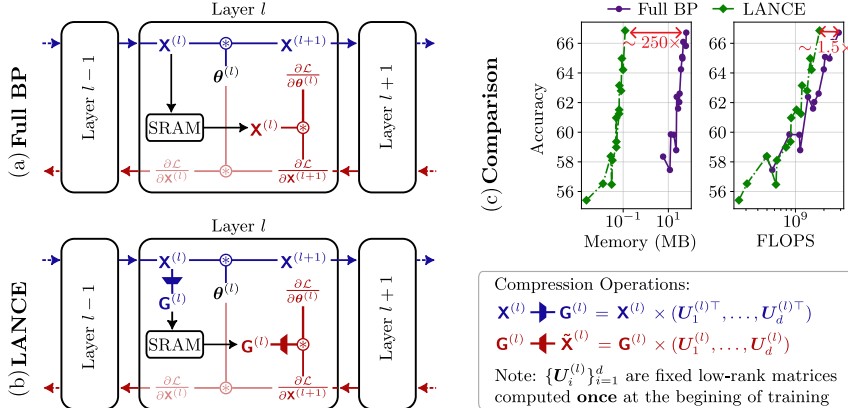

Figure 1: Overview of LANCE for on-device training. (a) In full backpropagation (BP), the entire activation tensor $\mathbf{X}^{(l)}$ must be stored in memory for computing gradients, leading to a large memory footprint. (b) LANCE replaces $\mathbf{X}^{(l)}$ with a compressed core tensor $\mathbf{G}^{(l)}$, obtained via one-shot HOSVD using fixed low-rank matrices $\{\boldsymbol{U}_i^{(l)}\}_{i=1}^d$ computed once at the beginning of training. Only $\mathbf{G}^{(l)}$ is stored, while the factors are reused during the backward pass. (c) Pareto comparisons show that LANCE reduces memory (SRAM) usage by up to $\sim 250\times$ and FLOPs by $\sim 1.5\times$ relative to vanilla BP, while maintaining accuracy.

Instead of storing the full tensor $\mathbf{X}^{(l)}$, we can approximate it with a compressed representation and reconstruct (or use) it during backpropagation. While classical singular value decomposition (SVD) is defined for matrices, activations are inherently higher-order (e.g., batch × channels × height × width), making tensor decompositions more suitable.

**Higher-Order Singular Value Decomposition (HOSVD).** HOSVD (or $N$-mode SVD) provides a principled way to decompose a tensor into mode-specific subspaces. Given an order-$d$ activation tensor $\mathbf{X} \in \mathbb{R}^{n_1 \times n_2 \times \cdots \times n_d}$, HOSVD expresses it as $\mathbf{X} = \mathbf{G} \times (\boldsymbol{U}_1, \boldsymbol{U}_2, \cdots, \boldsymbol{U}_d)$, where $\mathbf{G} \in \mathbb{R}^{r_1 \times r_2 \times \cdots \times r_d}$ is the core tensor, $\boldsymbol{U}_i \in \mathbb{R}^{n_i \times r_i}$ are orthonormal factor matrices for each mode $i$, and $\times$ denotes the tensor–matrix product. The ranks $\{r_i\}_{i=1}^d$ control the compression level, with $r_i \ll n_i$ in the low-rank regime. To compute HOSVD, the tensor $\mathbf{X}$ is *unfolded* along each mode $i$ into a matrix $\boldsymbol{X}_i \in \mathbb{R}^{n_i \times \prod_{j \neq i} n_j}$, followed by a matrix SVD $\boldsymbol{X}_i = \boldsymbol{U}_i \boldsymbol{\Sigma}_i \boldsymbol{V}_i^\top$. Truncating to the top-$r_i$ singular vectors yields $\boldsymbol{U}^{(i)}$, which captures the dominant subspace of mode $i$. Thus, HOSVD produces a compact multi-linear approximation of $\mathbf{X}$ that preserves most of its structure while discarding redundancy.

## 2.2 PROPOSED METHOD: LANCE

To address the challenges of training DNNs on low-power devices, we propose LANCE, a one-shot HOSVD-based low-rank activation compression method for efficient on-device learning. LANCE directly targets the primary memory bottleneck in backpropagation: the need to store intermediate activations $\mathbf{X}^{(l)}$ from hidden layers. By projecting activations into a compact low-rank subspace, LANCE reduces both the memory required to store $\mathbf{X}^{(l)}$ and the number of computations needed to backpropagate through them.

Formally, given projection matrices $\{\boldsymbol{U}_i\}_{i=1}^d$ obtained via HOSVD, each activation tensor $\mathbf{X}^{(l)} \in \mathbb{R}^{n_1 \times n_2 \times \cdots \times n_d}$ is compressed into a smaller core tensor $\mathbf{G}^{(l)} \in \mathbb{R}^{r_1 \times r_2 \times \cdots \times r_d}$:

$$\mathbf{G}^{(l)} = \mathbf{X}^{(l)} \times (\boldsymbol{U}_1^{(l)\top}, \boldsymbol{U}_2^{(l)\top}, \cdots, \boldsymbol{U}_d^{(l)\top}). \tag{1}$$

The compressed $\mathbf{G}^{(l)}$ is saved instead of the full activation and reused during backpropagation. Unlike prior works that recompute low-rank decompositions every iteration, LANCE computes the projection matrices only once at the beginning of fine-tuning and reuses them across all epochs. This one-shot approach avoids expensive per-iteration SVDs and is particularly well-suited for energy-

---

**Algorithm 1** LANCE: Low-Rank Activation Compression for On-device Continual Learning

---

**Require:** Pretrained network $f(\mathbf{x}; \boldsymbol{\theta})$ with layers $l=1,\ldots,L$; calibration batches $\{\mathcal{B}_j\}_{j=1}^N$; energy thresholds $\varepsilon$ (LANCE) and $\varepsilon_{\mathrm{CL}}$ (CL); *(CL only)* memory $\boldsymbol{M}^{t-1}$ from previous task

1: *Notation: we omit layer superscripts for readability; all steps are applied per layer.*

2: **Phase I: One-Shot Subspace Calibration (offline)**

3: **for** each mode $i \in \{1,\ldots,d\}$ **do**

4:     Estimate covariance $\boldsymbol{B}_i$ from calibration activations via (2)

5:     **if** $i=d$ **then**

6:         $\hat{\boldsymbol{B}}_d \leftarrow (\boldsymbol{I} - \boldsymbol{M}^{t-1}\boldsymbol{M}^{t-1\top})\,\boldsymbol{B}_d$         // *For non-CL or $t=1$, $\boldsymbol{M}^{t-1}$ is a zero matrix*

7:         Compute SVD, $\hat{\boldsymbol{B}}_d = \hat{\boldsymbol{U}}_d\hat{\boldsymbol{\Sigma}}_d\hat{\boldsymbol{U}}_d^\top$, and $\boldsymbol{B}_d = \boldsymbol{U}_d\boldsymbol{\Sigma}_d\boldsymbol{U}_d^\top$

8:         Retain top $r_d$ columns of $\hat{\boldsymbol{U}}_d$ satisfying (5); set $\boldsymbol{U}_d \leftarrow \hat{\boldsymbol{U}}_d[:, 1:r_d]$

9:     **else**

10:         Compute SVD $\boldsymbol{B}_i = \boldsymbol{U}_i\boldsymbol{\Sigma}_i\boldsymbol{U}_i^\top$

11:         Retain top $r_i$ columns of $\boldsymbol{U}_i$ satisfying (3)

12:     **end if**

13: **end for**

14: **Phase II: Fine-Tuning with Low-Rank Activations (on-device)**

15: **for** each minibatch $\mathcal{B}$ **do**

16:     Forward: project $\boldsymbol{X}^{(l)}$ to cores $\mathbf{G}^{(l)} = \mathbf{X}^{(l)} \times (\boldsymbol{U}_1^\top, \ldots, \boldsymbol{U}_d^\top)$; store $\mathbf{G}^{(l)}$ only

17:     Backward: compute $\nabla_{\boldsymbol{\theta}^{(l)}}\mathcal{L}$ using $\mathbf{G}^{(l)} \times (\boldsymbol{U}_1, \ldots, \boldsymbol{U}_d)$; update $\boldsymbol{\theta}^{(l)}$

18: **end for**

19: **Phase III (CL only): Memory Construction & Update (offline, after task $t$)**

20: Estimate last-mode covariance $\boldsymbol{B}_d^t$ from $N_{\mathrm{CL}}$ calibration mini-batches as in (2)

21: Compute $\hat{\boldsymbol{B}}_d^t \leftarrow (\boldsymbol{I} - \boldsymbol{M}^{t-1}\boldsymbol{M}^{t-1\top})\,\boldsymbol{B}_d^t$, then SVD on $\hat{\boldsymbol{B}}_d^t$ and $\boldsymbol{B}_d^t$

22: Select $r_d$ via (5); and update memory: $\boldsymbol{M}^t \leftarrow \mathrm{orth}\big([\boldsymbol{M}^{t-1} \ \ \hat{\boldsymbol{U}}_{d,[:,1:r_d]}^t]\big)$

---

constrained devices. The method, summarized in Algorithm 1, has two phases: (1) a one-shot subspace calibration performed offline, and (2) on-device fine-tuning with low-rank activations, which projects activations into low-rank subspaces during the forward pass and computes gradients during the backward pass using the compressed activations (Fig. 1b). Each step is detailed below.

**Phase I: One-Shot Subspace Calibration (offline).**  We estimate the subspace for each hidden layer $l$ using activations from $N$ calibration mini-batches passed through the pretrained model. For each mode $i$, we maintain a running covariance estimate:

$$\boldsymbol{B}_i[t] \;=\; \tfrac{1}{t}\Big((t-1)\boldsymbol{B}_i[t-1] \;+\; \tfrac{1}{\prod_{j\neq i} n_j}\boldsymbol{X}_i^{(l)}\boldsymbol{X}_i^{(l)\top}\Big), \tag{2}$$

with $\boldsymbol{B}_i[0]=0$, where $\boldsymbol{X}_i^{(l)}$ is the mode-$i$ unfolding of $\mathbf{X}^{(l)}$. After $N$ batches, we compute an SVD decomposition, $\boldsymbol{B}_i = \boldsymbol{U}_i\boldsymbol{\Sigma}_i\boldsymbol{U}_i^\top$. We then truncate to the top-$r_i$ singular vectors according to an energy threshold $\varepsilon$:

$$\frac{\sum_{j=1}^{r_i}\sigma_j}{\sum_{j=1}^{n_i}\sigma_j} \;\geq\; \varepsilon, \tag{3}$$

where $\sigma_j$ is the $j$-th singular value of $\boldsymbol{B}_i$. This ensures that the subspace captures at least an $\varepsilon$ fraction of activation variance. Because this decomposition is computed only once (before fine-tuning) and amortized across training, its cost is negligible relative to repeated decompositions.

**Phase II: Fine-Tuning with Low-Rank Activations (on-device).**  During fine-tuning with BP (Fig. 1b), we distinguish the forward and backward passes.

*Forward pass.* Each activation $\mathbf{X}^{(l)}$ is projected into the learned low-rank subspace using the pre-computed matrices $\{\boldsymbol{U}_i^{(l)}\}_{i=1}^d$, following (1) to obtain a core tensor $\mathbf{G}^{(l)}$. Only the compact tensor $\mathbf{G}^{(l)}$ and the fixed $\{\boldsymbol{U}_i^{(l)}\}_{i=1}^d$ need to be stored for backpropagation. This reduces the memory footprint from $\mathcal{O}(\prod_{i=1}^d n_i)$ to $\mathcal{O}(\prod_{i=1}^d r_i + \sum_{i=1}^d n_i r_i)$ per activation.

*Backward pass.* The gradient with respect to the weights is computed using the compressed activations. Given the gradient of the loss with respect to the layer output, $\nabla_{\mathbf{X}^{(l+1)}}\mathcal{L}$, we reconstruct the contribution with respect to the layer's parameters $\boldsymbol{\theta}^{(l)}$ by reversing the projections:

$$\nabla_{\boldsymbol{\theta}^{(l)}}\tilde{\mathcal{L}} \;=\; \frac{\partial\mathcal{L}^\top}{\partial\mathbf{X}^{(l+1)}}\left(\mathbf{G}^{(l)} \times (\boldsymbol{U}_1^{(l)}, \cdots, \boldsymbol{U}_d^{(l)})\right). \tag{4}$$

Because $\{\boldsymbol{U}_i^{(l)}\}_{i=1}^d$ are orthonormal, this operation preserves gradient directions up to the truncation error introduced by the low-rank approximation. We prove that the approximated gradient $-\nabla_{\boldsymbol{\theta}^{(l)}}\tilde{\mathcal{L}}$ is a descent direction and that the overall method converges in Sec. 2.4. Finally, gradients with respect to activations, $\nabla_{\mathbf{X}^{(l)}}\mathcal{L}$, are computed as usual by automatic differentiation; the compression only affects which forward tensors are stored (cores $\mathbf{G}^{(l)}$ vs. full $\mathbf{X}^{(l)}$), not the functional form of the backward.

## 2.3 LANCE FOR ON-DEVICE CONTINUAL LEARNING

We extend LANCE to the continual learning (CL) setting, where tasks arrive sequentially $t = 1, 2, \ldots, T$. After completing task $t$, we retain for each layer a compact set of *important directions* on the last mode (the input dimension of the layer's weights), denoted by $\boldsymbol{M}^t$. When calibrating the HOSVD for the next task $t+1$, the new columns of $\boldsymbol{U}_d$ (the mode-$d$ factors) are chosen to lie strictly in the *null space* of $\boldsymbol{M}^t$. This ensures that task-$t+1$ learns from directions orthogonal to those already used by previous tasks, thereby preventing interference and mitigating forgetting.

For clarity, we omit the explicit layer superscript in the following notation; all operations (memory construction, calibration, and updates) are applied independently at each layer.

**Phase III: Memory construction and update.** At the end of training on task $t$, we estimate the subspace $\boldsymbol{M}^t$ by accumulating the covariance of activations along the last mode, $\boldsymbol{B}_d^t$, as in (2) using $N_{\mathrm{CL}}$ calibration mini-batches. An SVD decomposition of $\boldsymbol{B}_d^t$ yields a basis of principal directions, and we retain the top $r_d$ components that capture at least an energy fraction $\varepsilon_{\mathrm{CL}}$. To ensure orthogonality with previously stored directions, we project $\boldsymbol{B}_d^t$ onto the complement of $\boldsymbol{M}^{t-1}$: $\hat{\boldsymbol{B}}_d^t = \boldsymbol{B}_d^t - \boldsymbol{M}^{t-1}\boldsymbol{M}^{t-1\top}\boldsymbol{B}_d^t$, where $\boldsymbol{M}^0$ is initialized as empty. We select the smallest $r_d$ such that:

$$\frac{\sum_{j=1}^{n_d}\sigma_j - \sum_{j=r_d}^{n_d}\hat{\sigma}_j}{\sum_{j=1}^{n_d}\sigma_j} \;\geq\; \varepsilon_{\mathrm{CL}}. \tag{5}$$

The corresponding top-$r_d$ eigenvectors form $\hat{\boldsymbol{U}}_d^t$, which are merged with the previous memory to update: $\boldsymbol{M}^t \leftarrow \mathrm{orth}\big([\boldsymbol{M}^{t-1}\ \hat{\boldsymbol{U}}_{d,[:,1:r_d]}^t]\big)$.

**Null-space constrained calibration.** For a new task $t+1$, we update the one-shot calibration of LANCE by incorporating the memory constraint only in mode-$d$; all other modes follow the procedure in Section 2.2. We compute the covariance $\boldsymbol{B}_d$ for the new task, remove contributions along $\boldsymbol{M}^t$, $\hat{\boldsymbol{B}}_d = \boldsymbol{B}_d - \boldsymbol{B}_d\boldsymbol{M}^t\boldsymbol{M}^{t\top}$, and select the top $r_d$ eigenvectors of $\hat{\boldsymbol{B}}_d$ that satisfy an energy threshold $\varepsilon$ similar to (5). The resulting low-rank factors used during fine-tuning are $\{\boldsymbol{U}_i^{(l)}\}_{i=1}^{d-1} \cup \hat{\boldsymbol{U}}_{d,[:,1:r_d]}^{(l)}$, where $\hat{\boldsymbol{U}}_{d,[:,1:r_d]}^{(l)}$ are the retained singular vectors of $\hat{\boldsymbol{B}}_d$. This guarantees that activations of the new task are compressed into a subspace orthogonal to those of previous tasks, thereby reducing catastrophic forgetting.

**Discussion.** Both memory update and null-space constrained calibration are performed *offline*, so they do not affect the efficiency of on-device fine-tuning. The method requires storing only the compact memory matrices $\{\boldsymbol{M}^{(l)}\}$ and applying inexpensive projections during calibration. By construction, interference with past tasks is eliminated along the stored subspaces, and forgetting is mitigated without replay buffers or large task-specific models. In contrast to prior gradient-projection methods (Saha et al., 2021; Saha & Roy, 2023; Lin et al., 2022b;c; Apolinario et al., 2025a), which explicitly use $\{\boldsymbol{M}^{(l)}\}$ for online gradient projection and must store such matrices in SRAM during training, our approach achieves continual learning naturally by projecting activations into fixed low-rank subspaces that lie in the null space of previous tasks, retaining LANCE's memory efficiency.

## 2.4 CONVERGENCE ANALYSIS

We provide a brief convergence analysis showing that LANCE yields valid descent directions and converges to projected stationary points. Full proofs are deferred to Appendix B.

**Assumption A1** (Orthogonal truncation). For every layer $l$ and mode $i$, the calibration bases $\boldsymbol{U}_i^{(l)}$ have orthonormal columns; hence $\mathsf{P}^{(l)} := \boldsymbol{U}_1^{(l)}\boldsymbol{U}_1^{(l)\top} \otimes \cdots \otimes \boldsymbol{U}_d^{(l)}\boldsymbol{U}_d^{(l)\top}$ is an orthogonal projector.

**Assumption A2** (Smoothness). The empirical loss $\mathcal{L}(\boldsymbol{\theta})$ is $L$-smooth.

**Assumption A3** (Layer linearity in inputs). Each trainable layer is linear in its input during back-propagation (dense or convolutional after unfolding). Nonlinearities are piecewise linear and treated as fixed locally.

**Assumption A4** (Energy capture). One-shot HOSVD retains a fraction $\varepsilon \in (0, 1]$ of activation energy per mode, inducing a bound on gradient leakage outside the retained input subspace (used in Proposition 2.3).

**Theorem 2.1** (Projected gradient & descent). *Under Assumption A1 and Assumption A3, for any layer $l$ with input projector $\mathsf{P}^{(l)}$, the LANCE weight gradient equals the right-Frobenius orthogonal projection of the full gradient: $\nabla_{\boldsymbol{W}}\mathcal{L}_{\mathrm{LANCE}} = \nabla_{\boldsymbol{W}}\mathcal{L}_{\mathrm{full}}\,\mathsf{P}^{(l)}$. Consequently, $\langle\nabla_{\boldsymbol{W}}\mathcal{L}_{\mathrm{LANCE}}, \nabla_{\boldsymbol{W}}\mathcal{L}_{\mathrm{full}}\rangle = \|\nabla_{\boldsymbol{W}}\mathcal{L}_{\mathrm{LANCE}}\|_F^2 \geq 0$, so $-\nabla_{\boldsymbol{W}}\mathcal{L}_{\mathrm{LANCE}}$ is a descent direction unless it vanishes.*

**Theorem 2.2** (Monotone decrease & projected stationarity). *Let $\mathsf{Q}$ be the block projector that right-multiplies each layer's weight gradient by its $\mathsf{P}^{(l)}$. Under Assumption A2, the LANCE update $\boldsymbol{\theta}_{k+1} = \boldsymbol{\theta}_k - \eta\,\mathsf{Q}\,\nabla\mathcal{L}(\boldsymbol{\theta}_k)$ with $\eta \in (0, 1/L]$ satisfies $\mathcal{L}(\boldsymbol{\theta}_{k+1}) \leq \mathcal{L}(\boldsymbol{\theta}_k) - \frac{\eta}{2}\|\mathsf{Q}\nabla\mathcal{L}(\boldsymbol{\theta}_k)\|_2^2$, hence $\|\mathsf{Q}\nabla\mathcal{L}(\boldsymbol{\theta}_k)\|_2 \to 0$ and every limit point $\boldsymbol{\theta}_\star$ obeys $\mathsf{Q}\nabla\mathcal{L}(\boldsymbol{\theta}_\star) = \mathbf{0}$.*

**Proposition 2.3** (Stationarity gap vs. truncation). *If along the iterates $\|(I - \mathsf{Q})\nabla\mathcal{L}(\boldsymbol{\theta})\|_2 \leq C\sqrt{1-\varepsilon}$ for some $C > 0$, then any limit point $\boldsymbol{\theta}_\star$ satisfies $\|\nabla\mathcal{L}(\boldsymbol{\theta}_\star)\|_2 \leq C\sqrt{1-\varepsilon}$.*

**Insight.** *Together, these results show that LANCE behaves like full BP restricted to the low-rank activation subspace: updates are valid descent directions, the loss decreases monotonically, and iterates converge to projected stationary points. The residual gap depends only on the energy threshold $\varepsilon$, so higher $\varepsilon$ yields solutions closer to true stationary points, while lower $\varepsilon$ trades accuracy for efficiency in a quantifiable way.*

## 3 EXPERIMENTAL EVALUATION

We evaluate LANCE in two complementary settings: (i) *single-task fine-tuning*, where a pretrained model is adapted to a target dataset on an edge device, and (ii) *continual learning*, where multiple tasks arrive sequentially and must be learned without catastrophic forgetting. Our experiments aim to answer the following questions:

**Q1** Does the one-shot subspace calibration preserve gradient fidelity in practice?

**Q2** How effective is LANCE at reducing activation memory and computational cost during on-device fine-tuning, while maintaining accuracy?

**Q3** Can LANCE leverage fixed low-rank subspaces to achieve competitive performance on continual learning benchmarks, while offering superior memory efficiency?

**Experimental Setup.** (i) Datasets: For single-task fine-tuning, we use CIFAR-10/100, Oxford-IIIT Pets, Flowers-102, and CUB-200, covering a range of dataset sizes and complexities. For continual learning, we evaluate on Split CIFAR-100 (20 tasks), Split MiniImageNet (20 tasks), and the 5-Datasets benchmark (CIFAR-10, MNIST, SVHN, Fashion-MNIST, and notMNIST), following common CL protocols. (ii) Metrics: We report the following metrics: classification accuracy on the target datasets; memory usage, measured as peak activation storage in MB (estimated in Appendix C); and computational cost in FLOPs. For continual learning benchmarks, we additionally report the average final accuracy over all tasks (ACC) and the Backward Transfer (BWT), which quantifies the forgetting of previously learned tasks when new tasks are introduced, as defined in Appendix A.2.3.

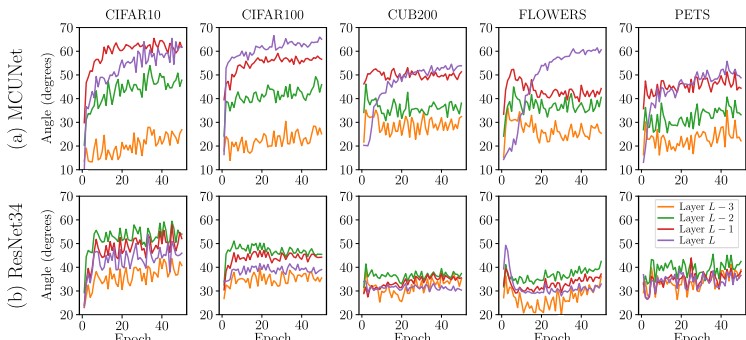

Figure 2: Gradient alignment between full BP and LANCE. We plot the angle between true gradients and LANCE-projected gradients across epochs for different fine-tuning tasks. LANCE consistently produces gradients within 70° of the true gradient, and angles stabilize as training progresses, indicating preserved descent directions.

Table 1: ImageNet-scale fine-tuning on Split ImageNet-1k. We pretrain each model on 500 ImageNet classes (Split A) and fine-tune on the remaining 500 classes (Split B).

| | ResNet18 | | | ResNet34 | | |
|---|---|---|---|---|---|---|
| Method | Acc ↑ | MB ↓ | TFLOPs ↓ | Acc ↑ | MB ↓ | TFLOPs ↓ |
| BP (2 layers) | 60.70 | 24.50 | 0.059 | 60.77 | 24.50 | 0.059 |
| BP (4 layers) | 64.28 | 61.25 | 0.090 | 64.21 | 49.00 | 0.118 |
| LANCE (2 layers) | 59.18 | 7.12 | 0.048 | 58.59 | 4.49 | 0.046 |
| LANCE (4 layers) | 62.20 | 16.59 | 0.075 | 62.66 | 10.79 | 0.093 |

## 3.1 RESULTS: SINGLE-TASK FINE-TUNING EFFICIENCY

We first evaluate LANCE on fine-tuning the last 2 or 4 layers of MCUNet (Lin et al., 2020), MobileNetV2 (Sandler et al., 2018), ResNet18 (He et al., 2016) and ResNet34 (He et al., 2016). All models are pretrained on ImageNet-1k (Krizhevsky et al., 2012). For LANCE, the offline Phase I calibration uses $N=100$ mini-batches with an energy threshold $\varepsilon=0.7$. Fine-tuning runs for 50 epochs.

**Answer to Q1.** To assess gradient fidelity, we compute the angle between LANCE gradients ($\nabla_{\theta}\mathcal{L}_{\text{LANCE}}$) and full BP gradients ($\nabla_{\theta}\mathcal{L}_{\text{BP}}$), normalized as $\nabla_{\theta}\mathcal{L}/||\nabla_{\theta}\mathcal{L}||_F$, on one mini-batch at the start of each epoch. Fig. 2 shows that LANCE gradients remain within $\sim70°$ of the full gradients, with angles stabilizing over time. This empirically validates Theorem 2.1, confirming that LANCE preserves gradient fidelity and maintains valid descent directions. This observation is consistent with the ablation in Sec. 3.3, where gradient fidelity remains stable across a wide range of compression ratios controlled by the energy threshold $\varepsilon$.

**Answer to Q2.** Across all datasets and models, LANCE reduces activation storage by up to $250\times$ relative to BP while maintaining competitive accuracy. Table 2 reports results on CIFAR-10/100 and Pets and CUB200. Compared to iterative low-rank methods such as ASI (Nguyen et al., 2025), or HOSVD (Nguyen et al., 2024) where a full SVD is recomputed at each step, LANCE achieves comparable compression with higher accuracy and similar FLOPs, confirming that repeated decompositions are unnecessary. Although BP attains higher accuracy when the same number of layers are fine-tuned, LANCE's extreme memory efficiency enables fine-tuning more layers under strict SRAM budgets. For example, fine-tuning 4 layers with LANCE reaches the accuracy of 2-layer BP while still fitting within microcontroller constraints. These results demonstrate that fixed low-rank subspaces retain sufficient representational power for efficient on-device fine-tuning.

**Scalability to ImageNet-Scale Fine-Tuning** To assess the scalability of LANCE to larger datasets and higher-capacity models, we conduct an ImageNet-scale experiment following the protocol used in prior work (Nguyen et al., 2025). We split ImageNet-1k into two disjoint subsets of 500 classes (Split A and Split B). A model is first pretrained on Split A and then fine-tuned on Split B using either full BP or LANCE. Table 1 reports results for ResNet18 and ResNet34 when fine-tuning the last 2 or 4 layers. The trends closely mirror those observed on smaller downstream datasets: LANCE

Table 2: Performance and efficiency comparison. Results are reported in terms of accuracy, memory usage, and TFLOPs. [†] denote the results from the respective original paper.

| Mdl. | Method | # Layers | CIFAR10 | | | CIFAR100 | | | Pets | | | CUB200 | | |
|---|---|---|---|---|---|---|---|---|---|---|---|---|---|---|
| | | | Acc ↑ | MB ↓ | TFLOPS ↓ | Acc ↑ | MB ↓ | TFLOPS ↓ | Acc ↑ | MB ↓ | TFLOPS ↓ | Acc ↑ | MB ↓ | TFLOPS ↓ |
| MCUNet | BP | 2 | 75.85 | 11.71 | 0.000 | 52.47 | 11.71 | 0.000 | 56.30 | 11.71 | 0.000 | 33.05 | 11.71 | 0.000 |
| | | 4 | 82.91 | 17.57 | 0.001 | 58.38 | 17.57 | 0.001 | 60.01 | 17.57 | 0.001 | 34.13 | 17.57 | 0.001 |
| | HOSVD | 2 | 70.91 | 0.04 | 1.930 | 48.00 | 0.05 | 1.930 | 56.11 | 0.06 | 1.93 | 31.22 | 0.06 | 1.93 |
| | | 4 | 75.63 | 0.12 | 2.590 | 52.34 | 0.10 | 2.590 | 57.89 | 0.17 | 2.59 | 31.72 | 0.14 | 2.59 |
| | LANCE | 2 | 70.64 | 0.04 | 0.000 | 47.57 | 0.04 | 0.000 | 57.26 | 0.06 | 0.000 | 31.44 | 0.05 | 0.000 |
| | | 4 | 76.23 | 0.09 | 0.000 | 52.59 | 0.09 | 0.000 | 58.05 | 0.14 | 0.000 | 30.87 | 0.13 | 0.000 |
| MobileNetV2 | BP | 2 | 91.72 | 30.62 | 0.01 | 73.43 | 30.62 | 0.01 | 89.94 | 30.62 | 0.01 | 64.37 | 30.62 | 0.01 |
| | | 4 | 92.23 | 57.42 | 0.01 | 74.69 | 57.42 | 0.01 | 90.32 | 57.42 | 0.01 | 65.86 | 57.42 | 0.01 |
| | HOSVD | 2 | 85.67 | 0.13 | 11.87 | 67.28 | 0.14 | 11.87 | 89.58 | 0.13 | 11.87 | 58.11 | 0.13 | 11.87 |
| | | 4 | 89.03 | 0.16 | 22.86 | 69.33 | 0.15 | 22.86 | 89.91 | 0.23 | 22.86 | 60.30 | 0.16 | 22.86 |
| | ASI [†] | 2 | 85.09 | 0.26 | 0.01 | 61.03 | 0.15 | 0.01 | 88.13 | 0.15 | 0.01 | 46.44 | 0.26 | 0.01 |
| | | 4 | 88.03 | 0.63 | 0.06 | 66.18 | 0.63 | 0.06 | 88.91 | 0.76 | 0.06 | 50.69 | 0.63 | 0.06 |
| | LANCE | 2 | 84.16 | 0.15 | 0.01 | 60.69 | 0.16 | 0.01 | 89.69 | 0.14 | 0.01 | 58.18 | 0.14 | 0.01 |
| | | 4 | 88.99 | 0.17 | 0.01 | 68.79 | 0.17 | 0.01 | 90.40 | 0.19 | 0.01 | 61.21 | 0.15 | 0.01 |
| ResNet18 | BP | 2 | 92.69 | 24.5 | 0.05 | 75.69 | 24.5 | 0.06 | 89.17 | 24.5 | 0.06 | 62.28 | 24.5 | 0.06 |
| | | 4 | 94.01 | 61.25 | 0.09 | 77.07 | 61.25 | 0.09 | 89.07 | 61.25 | 0.09 | 60.80 | 61.25 | 0.09 |
| | HOSVD | 2 | 92.17 | 0.71 | 6.12 | 74.57 | 0.72 | 6.12 | 89.47 | 0.92 | 6.12 | 60.02 | 0.73 | 6.12 |
| | | 4 | 93.16 | 1.16 | 15.56 | 75.35 | 1.21 | 15.56 | 88.71 | 1.77 | 15.56 | 59.26 | 1.41 | 15.56 |
| | ASI [†] | 2 | 90.77 | 1.24 | 0.04 | 69.70 | 0.90 | 0.04 | 88.75 | 2.01 | 0.04 | 56.77 | 1.51 | 0.04 |
| | | 4 | 91.90 | 1.89 | 0.06 | 71.93 | 1.61 | 0.06 | 88.44 | 4.34 | 0.07 | 55.73 | 3.50 | 0.07 |
| | LANCE | 2 | 91.8 | 0.75 | 0.03 | 74.09 | 0.77 | 0.03 | 89.26 | 0.94 | 0.03 | 59.87 | 0.73 | 0.03 |
| | | 4 | 92.89 | 1.11 | 0.05 | 75.7 | 01.16 | 0.05 | 89.12 | 1.78 | 0.05 | 59.50 | 1.36 | 0.05 |
| ResNet34 | BP | 2 | 93.48 | 24.50 | 0.06 | 76.19 | 24.50 | 0.06 | 91.22 | 24.50 | 0.06 | 64.10 | 24.50 | 0.06 |
| | | 4 | 94.37 | 49.00 | 0.12 | 77.50 | 49.00 | 0.12 | 91.60 | 49.00 | 0.12 | 64.22 | 49.00 | 0.12 |
| | HOSVD | 2 | 92.70 | 0.42 | 6.12 | 75.31 | 0.45 | 6.12 | 90.89 | 0.48 | 6.12 | 60.82 | 0.38 | 6.12 |
| | | 4 | 93.80 | 0.92 | 12.25 | 76.01 | 0.95 | 12.25 | 91.11 | 1.10 | 12.25 | 62.28 | 0.84 | 12.25 |
| | ASI [†] | 2 | 90.09 | 0.49 | 0.03 | 69.66 | 0.44 | 0.03 | 91.09 | 0.81 | 0.04 | 58.77 | 0.60 | 0.03 |
| | | 4 | 91.54 | 1.24 | 0.07 | 70.60 | 1.00 | 0.07 | 91.41 | 2.05 | 0.07 | 58.85 | 1.58 | 0.07 |
| | LANCE | 2 | 92.62 | 0.46 | 0.03 | 74.89 | 0.51 | 0.03 | 90.70 | 0.45 | 0.03 | 62.35 | 0.39 | 0.03 |
| | | 4 | 93.56 | 1.01 | 0.06 | 75.94 | 1.02 | 0.06 | 90.92 | 1.05 | 0.06 | 63.01 | 0.80 | 0.06 |

achieves large reductions in activation memory (up to ∼4×) and maintains accuracy within ∼1–2% of full BP, with only modest differences in training FLOPs. These results demonstrate that LANCE remains effective and memory-efficient even at ImageNet scale, confirming that the one-shot low-rank subspace generalizes well to large, high-variance visual distributions.

Table 3: Performance comparison on continual image classification datasets using multi-head networks. Accuracy and BWT (mean ± std) are reported over five trials. [†] denotes the results taken from Saha et al. (2021) and [‡] denote the results from the respective original papers.

| Method | Split CIFAR100 | | | Split MiniImageNet | | | 5-Datasets | | |
|---|---|---|---|---|---|---|---|---|---|
| | ACC (%) ↑ | BWT (%) ↑ | Mem (MB) ↓ | ACC (%) ↑ | BWT (%) ↑ | Mem (MB) ↓ | ACC (%) ↑ | BWT (%) ↑ | Mem (MB) ↓ |
| Multitask [†] | 79.58 ± 0.54 | – | – | 69.46 ± 0.62 | – | – | 91.54 ± 0.28 | – | – |
| EWC [†] | 68.80 ± 0.88 | -2 ± 1 | – | 52.01 ± 2.53 | -12 ± 3 | – | 88.64 ± 0.26 | -4 ± 1 | – |
| HAT [†] | 72.06 ± 0.50 | 0 ± 0 | – | 59.78 ± 0.57 | -3 ± 0 | – | 91.32 ± 0.18 | -1 ± 0 | – |
| A-GEM [†] | 63.98 ± 1.22 | -15 ± 2 | – | 57.24 ± 0.72 | -12 ± 1 | – | 84.04 ± 0.33 | -12 ± 1 | – |
| ER_Res [†] | 71.73 ± 0.63 | -6 ± 1 | – | 58.94 ± 0.85 | -7 ± 1 | – | 80.31 ± 0.22 | -4 ± 0 | – |
| TRGP+SD [‡] | 75.5 ± 0.35 | -0.96 ± 0.09 | – | 65.8 ± 0.16 | -0.49 ± 0.08 | – | – | – | – |
| LMSP [‡] | 74.21 | +0.94 | – | 64.2 | +1.55 | – | 93.78 | +0.07 | – |
| GPM [†] | 72.48 ± 0.54 | -0.9 | 22.27 | 60.41 ± 0.61 | -0.9 | 127.37 | 91.22 ± 0.20 | -1.0 | 70.33 |
| TRGP [‡] | 74.64 ± 0.32 | -0.9 ± 0.01 | 78.86 | 61.78 ± 0.60 | -0.5 ± 0.60 | 543.22 | 93.56 ± 0.10 | -0.04 ± 0.01 | 181.57 |
| CUBER [‡] | 75.54 ± 0.22 | +0.13 ± 0.08 | 326.79 | 62.67 ± 0.74 | +0.23 ± 0.15 | 604.61 | 93.48 ± 0.10 | -0.00 ± 0.02 | 196.91 |
| SGP [‡] | 76.05 ± 0.43 | -1 | 22.27 | 62.83 ± 0.33 | -1 | 127.37 | – | – | – |
| CODE-CL [‡] | 77.21 ± 0.32 | -1.1 ± 0.28 | 33.80 | 71.16 ± 0.32 | -1.1 ± 0.3 | 283.82 | 93.51 ± 0.13 | -0.11 ± 0.01 | 116.72 |
| **LANCE** | 71.52 ± 0.27 | -0.17 ± 0.34 | 2.32 | 59.68 ± 1.17 | -0.99 ± 0.78 | 32.96 | 90.76 ± 0.31 | -1.02 ± 0.15 | 34.96 |

## 3.2 RESULTS: CONTINUAL LEARNING

We compare LANCE against a broad set of continual learning baselines: regularization-based methods (EWC (Kirkpatrick et al., 2017), HAT (Serrà et al., 2018)), replay-based approaches (A-GEM (Chaudhry et al., 2019b), ER_Res (Chaudhry et al., 2019a)), and projection-based methods (GPM (Saha et al., 2021), TRGP (Lin et al., 2022b), CUBER (Lin et al., 2022c), SGP (Saha & Roy, 2023), SD (Zhao et al., 2023), LMSP (Shang et al., 2025), CODE-CL (Apolinario et al., 2025a)). For reference, we also report multitask performance, which serves as an upper bound where all

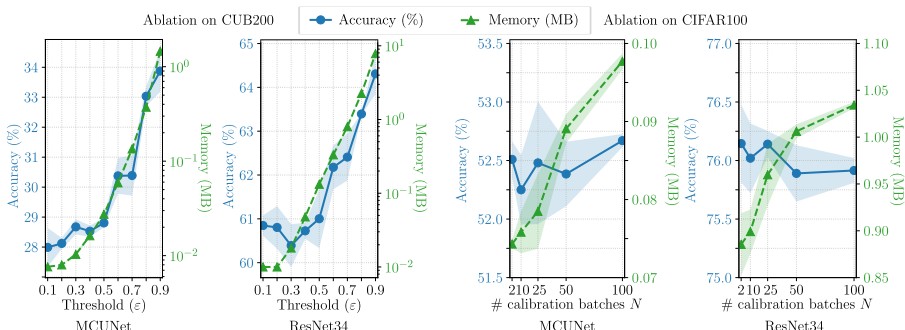

Figure 3: Ablation studies of LANCE. (Left) Effect of the energy threshold $\varepsilon$ on CUB-200 using MCUNet and ResNet34. Accuracy improves steadily as $\varepsilon$ increases, but memory usage grows exponentially, illustrating the trade-off between accuracy and compression. (Right) Effect of the number of calibration batches $N$ on CIFAR-100. Accuracy remains stable even for very small $N$, while memory increases with larger calibration sets. These results show that stable subspaces can be obtained with as few as $N{=}2$ calibration batches, and practical choices of $\varepsilon$ (e.g., 0.7) provide a good balance between accuracy and memory.

tasks are trained jointly. We follow the same experimental setup as Apolinario et al. (2025a): a 5-layer AlexNet for Split CIFAR-100, and a reduced ResNet18 for Split MiniImageNet and the 5-Datasets benchmark. During the first task, models are trained without constraints using full BP. For subsequent tasks, the model from the previous task serves as initialization, and LANCE applies its subspace-constrained training.

**Answer to Q3.** LANCE achieves accuracy competitive with full-rank gradient projection methods such as GPM and CODE-CL, while using drastically less memory. As summarized in Table 3, on Split CIFAR-100 LANCE reaches $\sim$71% average accuracy (ACC) while reducing activation storage by more than $10\times$ compared to GPM. Similar trends are observed on Split MiniImageNet and the 5-Datasets benchmark, where LANCE maintains stable performance without rehearsal memory. These results confirm that fixed low-rank subspaces are sufficient to retain representational capacity across tasks, while providing substantial memory savings for resource-constrained devices.

### 3.3 ABLATIONS AND ANALYSIS

We conduct ablations on two key hyperparameters: (i) the energy threshold $\varepsilon$ (evaluated on CUB-200), and (ii) the number of calibration batches $N$ (evaluated on CIFAR-100 with $\varepsilon{=}0.7$). Results for MCUNet and ResNet34 are shown in Fig. 3. As expected, higher $\varepsilon$ increases accuracy but also grows memory usage exponentially, highlighting the need to balance compression and accuracy according to device memory budgets. For calibration size $N$, accuracy remains stable across a wide range, while memory usage grows with $N$. Remarkably, even $N{=}2$ calibration batches are sufficient to yield stable subspaces, underscoring the robustness of the one-shot calibration phase.

**Gradient fidelity under compression.** We additionally evaluate how different compression ratios affect gradient fidelity by measuring the angle between full-BP gradients and LANCE-projected gradients across several values of $\varepsilon$. As expected, more aggressive compression (lower $\varepsilon$) leads to larger angles, but the degradation is smooth and controlled. Even at the strongest compression settings, reducing activation storage by up to two orders of magnitude, LANCE maintains gradient alignment within $\sim 70°$ of full BP, indicating that the one-shot subspace preserves the dominant descent directions required for stable optimization.

**Related Work.** The memory cost of storing intermediate activations has long been recognized as the central bottleneck for training on resource-constrained devices (Cai et al., 2020; Lin et al., 2022a). Prior work has reduced this cost through recomputation (e.g., checkpointing (Chen et al., 2016)), architectural modifications (e.g., reversible networks (Gomez et al., 2017)), or lossy compression and sparsity (Chen et al., 2021; Barley & Fröning, 2023; Jiang et al., 2022), but often at the expense of extra computation, accuracy, or hardware complexity. A complementary line of work exploits low-rank structure in activations (Nguyen et al., 2024; 2025). Closest to our method

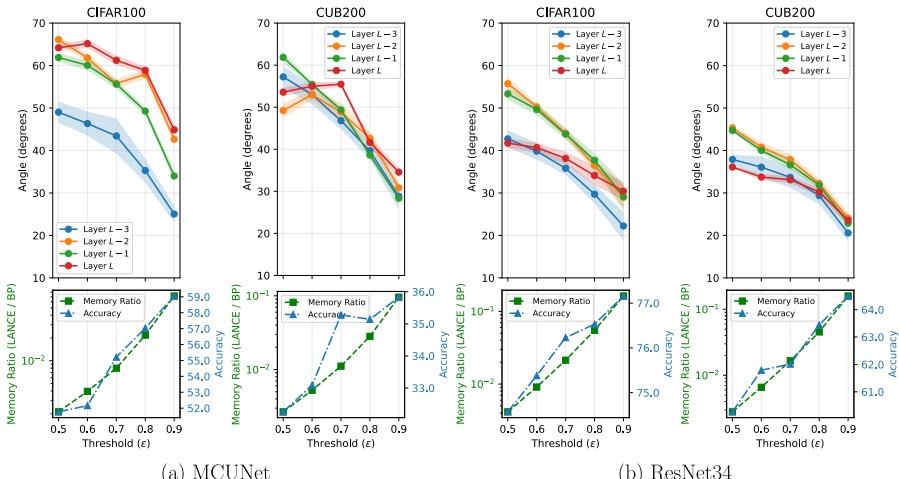

(a) MCUNet          (b) ResNet34

Figure 4: ==Ablation of gradient fidelity across compression ratios. We vary the energy threshold $\varepsilon$ controlling the effective memory compression of LANCE and report the mean and standard deviation of the angle between full-BP gradients and LANCE-projected gradients over the final 10 epochs (out of 50) for CIFAR-100 and CUB-200 using MCUNet and ResNet34. As $\varepsilon$ decreases, compression becomes more aggressive and gradient alignment degrades accordingly. Nevertheless, even under memory reductions of up to two orders of magnitude, LANCE maintains gradient directions within $\sim 70°$ of full BP, indicating preserved descent directions despite strong compression.==

is ASI (Nguyen et al., 2025), which incrementally updates subspaces during training; in contrast, LANCE performs a one-shot HOSVD at initialization and reuses the resulting subspaces throughout fine-tuning, avoiding repeated decompositions. ==Recent CL work such as== LMSP (Shang et al., 2025) ==also uses low-rank representations but focuses on reducing the computational cost of gradient projection, still requiring multiple per-task low-rank memory components; by contrast, LANCE targets the dominant activation-memory bottleneck directly.== This simple design not only reduces compute and hardware overhead but also enables a continual learning extension: fixed subspaces naturally support null-space allocation across tasks without storing large task-specific matrices. System-level methods such as TinyTL (Cai et al., 2020), MCUNet (Lin et al., 2020), and the 256KB training engine (Lin et al., 2022a) co-design architectures and update rules to fit extreme SRAM budgets, often by freezing layers or pruning gradient paths. LANCE is complementary to these approaches: instead of restricting the optimization, it preserves full backpropagation while directly compressing activations. Finally, in continual learning, regularization methods (Kirkpatrick et al., 2017; Serrà et al., 2018), replay-based approaches (Chaudhry et al., 2019b;a), and gradient-projection methods (Saha et al., 2021; Lin et al., 2022b;c; Saha & Roy, 2023; Apolinario et al., 2025a; Zhao et al., 2023) all mitigate forgetting by controlling parameter updates or storing exemplars. LANCE differs by operating directly in the activation subspace: new tasks are assigned to the orthogonal complement of previously used directions during calibration. This yields performance competitive with gradient-projection methods while requiring significantly less memory, and without rehearsal buffers.

## 4 CONCLUSION

We introduced LANCE, a one-shot low-rank activation compression framework for efficient on-device learning. By calibrating reusable activation subspaces with a single HOSVD, LANCE eliminates repeated decompositions and reduces both memory and computational cost. Experiments show that LANCE achieves up to $250\times$ memory savings while maintaining accuracy close to full backpropagation across diverse datasets and models. Crucially, the fixed low-rank subspaces naturally extend to continual learning, where LANCE matches the performance of gradient projection methods at a fraction of their memory footprint. Together, these results establish LANCE as a practical and scalable approach for enabling fine-tuning and continual learning on resource-constrained edge devices.

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

# A   DATASETS, MODELS AND HYPERPARAMETERS

This section provides details on the architecture of all models used in this work, the dataset statistics, the hyperparameters for each experiment, and the compute resources employed.

**Reported metrics.**   For single-task fine-tuning we report top-1 accuracy on the test split, peak activation memory in MB (estimated from tensors retained for the backward pass), and training FLOPs.[1]   For continual learning experiments (Section 2.3), we additionally report ACC and BWT following standard CL definitions.

## A.1   FOR SINGLE-TASK FINE-TUNING EXPERIMENTS

### A.1.1   DATASETS

We evaluate on five image classification datasets spanning scale and difficulty:  CIFAR-10 (Krizhevsky, 2009), CIFAR-100 (Krizhevsky, 2009), Oxford-IIIT Pets (Parkhi et al., 2012), Flowers-102 (Nilsback & Zisserman, 2008), and CUB-200 (Wah et al., 2023). We use the official training/test splits provided by each dataset and do not carve out an additional validation set. For all datasets, images are resized to match the native input resolution of the chosen architecture (e.g., $224 \times 224$ for ImageNet-pretrained models), and standard data augmentation is applied consisting of random horizontal flipping.

### A.1.2   MODEL ARCHITECTURES

For single-task experiments we use four pretrained models: ResNet18, ResNet34 (He et al., 2016), MobileNetV2 (Sandler et al., 2018), and MCUNet (Lin et al., 2020). All of them are initialized from ImageNet-1k pretrained weights. For each model, we unfreeze the last $N \in \{2, 4\}$ convolutional layers and fine-tune them while the rest of the network remains frozen.

### A.1.3   TRAINING PROTOCOL

Training runs for 50 epochs with stochastic gradient descent (SGD), learning rate 0.05, batch size 128, and without weight decay. For LANCE, we allocate $N=100$ calibration mini-batches to estimate the activation covariance (unless otherwise specified) and compute truncated subspaces. The energy threshold $\varepsilon$ for subspace retention 0.7. During training, activations of the selected layers are projected into the fixed low-rank cores, which are the only tensors retained for the backward pass.

### A.1.4   HYPERPARAMETERS

The hyperparameters used in our single-task fine-tuning experiments are summarized in Table 4.

Table 4: Hyperparameters used in single-task fine-tuning experiments.

| Hyperparameter | Value |
|---|---|
| Epochs | 50 |
| Optimizer | SGD |
| Learning rate | 0.05 |
| Batch size | 128 |
| Weight decay | 0 |
| # Unfrozen layers | 2 or 4 |
| # Calibration batches ($N$) | 100 |
| LANCE threshold $\varepsilon$ | 0.7 |

Table 5: 5-Datasets statistics.

| Dataset | CIFAR10 | MNIST | SVHN | Fashion MNIST | notMNIST |
|---|---|---|---|---|---|
| Number of classes | 10 | 10 | 10 | 10 | 10 |
| Training samples | 47500 | 57000 | 69595 | 57000 | 16011 |
| Validation samples | 2500 | 3000 | 3662 | 3000 | 842 |
| Test samples | 10000 | 10000 | 26032 | 10000 | 1873 |

Table 6: Split CIFAR100 and Split miniImageNet datasets statistics.

| Dataset | Split CIFAR100 | Split miniImageNet |
|---|---|---|
| Number of tasks ($T$) | 10 | 20 |
| Sample dimensions | $3 \times 32 \times 32$ | $3 \times 84 \times 84$ |
| Number of classes per task | 10 | 5 |
| Training samples per task | 4750 | 2375 |
| Validation samples per task | 250 | 125 |
| Test samples per task | 1000 | 500 |

## A.2  FOR CONTINUAL LEARNING EXPERIMENTS

### A.2.1  MODEL ARCHITECTURE

We employ two neural network architectures in our experiments: an AlexNet-inspired model following Serrà et al. (2018), and a Reduced ResNet18 as described in Lopez-Paz & Ranzato (2017). The AlexNet-like network integrates batch normalization (BN) layers after every convolutional and fully connected layer except the final classifier. BN parameters are updated only during the first task and kept fixed thereafter. Its architecture consists of three convolutional layers with 64, 128, and 256 filters, using kernel sizes of $4 \times 4$, $3 \times 3$, and $2 \times 2$, respectively, each followed by a $2 \times 2$ max-pooling operation. Two fully connected layers with 2048 units each are appended. ReLU activations are applied throughout, and dropout is used with rates of 0.2 for the first two layers and 0.5 for the rest. The Reduced ResNet18 follows the setup in Saha et al. (2021). For Split miniImageNet tasks, its first convolutional layer uses stride 2, whereas for the 5-Datasets benchmark, stride 1 is used. Across all experiments, models are trained with cross-entropy loss.

### A.2.2  DATASET STATISTICS

Tables 6 and 5 summarize the datasets employed in our continual learning benchmarks. We adopt the same train/test splits as in Saha et al. (2021); Lin et al. (2022b); Saha & Roy (2023); Apolinario et al. (2025a). For the 5-Datasets benchmark, grayscale samples are duplicated across three channels to match RGB input requirements. All images are resized to $32 \times 32$, yielding an effective input dimension of $3 \times 32 \times 32$.

### A.2.3  METRICS

Following standard practice in prior continual learning works (Lopez-Paz & Ranzato, 2017; Saha et al., 2021; Lin et al., 2022b; Saha & Roy, 2023; Apolinario et al., 2025a), we evaluate performance using two metrics: the average final accuracy across all tasks (ACC) and the Backward Transfer (BWT), which captures the extent to which learning new tasks affects performance on previously learned ones. They are defined as:

$$\text{ACC} = \sum_{i=1}^{T} \frac{A_{T,i}}{T} \; ; \text{BWT} = \sum_{i=1}^{T-1} \frac{A_{T,i} - A_{i,i}}{T - 1},$$
(6)

where $T$ denotes the total number of tasks, and $A_{j,i}$ represents the accuracy on the $i$-th task after the model has sequentially trained up to the $j$-th task ($i \leq j$).

---

[1]Peak activation memory and FLOPs formulas are detailed in Appendix C.

Table 7: List of hyperparameters used in Continual Learning experiments.

| Dataset | Split CIFAR100 | Split miniImageNet | 5-Datasets |
|---|---|---|---|
| Optimizer | SGD | SGD | SGD |
| Learning rate ($\eta$) | 0.01 | 0.1 | 0.1 |
| Batch size ($b$) | 64 | 64 | 64 |
| Min. learning rate ($\eta_{th}$) | $10^{-5}$ | $10^{-5}$ | $10^{-3}$ |
| Learning rate decay factor | 1/2 | 1/2 | 1/3 |
| Patience | 6 | 6 | 5 |
| Number of epochs ($E$) | 200 | 100 | 100 |
| Energy Threshold ($\varepsilon$) | 0.9 | 0.9 | 0.9 |
| CL Threshold ($\varepsilon_{\text{CL}}$) | 0.95 | 0.985 | 0.97 |
| # Calibration batches ($N$) | 100 | 100 | 100 |
| # batches for memory retention ($N_{\text{CL}}$) | 10 | 10 | 10 |

### A.2.4 HYPERPARAMETERS

The hyperparameters used in our experiments are detailed in Table 7.

### A.3 COMPUTE RESOURCES

All experiments were conducted on a shared internal Linux server equipped with an AMD EPYC 7502 32-Core Processor, 504 GB of RAM, and four NVIDIA A40 GPUs, each with 48 GB of GDDR6 memory. Additionally, code was implemented using Python 3.9 and PyTorch 2.2.1 with CUDA 11.8.

## B PROOFS FOR SECTION 2.4

**Theorem 2.1** (Projected gradient & descent). *Under Assumption A1 and Assumption A3, for any layer $l$ with input projector $\mathsf{P}^{(l)}$, the LANCE weight gradient equals the right-Frobenius orthogonal projection of the full gradient: $\nabla_{\boldsymbol{W}} \mathcal{L}_{\text{LANCE}} = \nabla_{\boldsymbol{W}} \mathcal{L}_{\text{full}} \mathsf{P}^{(l)}$. Consequently, $\langle \nabla_{\boldsymbol{W}} \mathcal{L}_{\text{LANCE}}, \nabla_{\boldsymbol{W}} \mathcal{L}_{\text{full}} \rangle = \|\nabla_{\boldsymbol{W}} \mathcal{L}_{\text{LANCE}}\|_F^2 \geq 0$, so $-\nabla_{\boldsymbol{W}} \mathcal{L}_{\text{LANCE}}$ is a descent direction unless it vanishes.*

*Proof.* Let $\delta$ denote the upstream error for the layer and let $\boldsymbol{x}$ be the unfolded input. For a linear (or unfolded convolutional) layer, the full gradient is $\nabla_{\boldsymbol{W}} \mathcal{L}_{\text{full}} = \delta \boldsymbol{x}^\top$. In LANCE, the only change (by design) is replacing $\boldsymbol{x}$ with $\mathsf{P}^{(l)} \boldsymbol{x}$, so

$$\nabla_{\boldsymbol{W}} \mathcal{L}_{\text{LANCE}} = \delta \, (\mathsf{P}^{(l)} \boldsymbol{x})^\top = \delta \, \boldsymbol{x}^\top \mathsf{P}^{(l)} = \nabla_{\boldsymbol{W}} \mathcal{L}_{\text{full}} \mathsf{P}^{(l)}.$$

Since $\mathsf{P}^{(l)}$ is an orthogonal projector (Assumption A1),

$$\langle \nabla_{\boldsymbol{W}} \mathcal{L}_{\text{LANCE}}, \nabla_{\boldsymbol{W}} \mathcal{L}_{\text{full}} \rangle = \text{tr}\Big( \big( \nabla_{\boldsymbol{W}} \mathcal{L}_{\text{full}} \mathsf{P}^{(l)} \big)^\top \nabla_{\boldsymbol{W}} \mathcal{L}_{\text{full}} \Big)$$

$$= \|\nabla_{\boldsymbol{W}} \mathcal{L}_{\text{full}} \mathsf{P}^{(l)}\|_F^2$$

$$= \|\nabla_{\boldsymbol{W}} \mathcal{L}_{\text{LANCE}}\|_F^2 \geq 0.$$

Thus $-\nabla_{\boldsymbol{W}} \mathcal{L}_{\text{LANCE}}$ is a descent direction unless it is zero. $\square$

**Theorem 2.2** (Monotone decrease & projected stationarity). *Let $\mathsf{Q}$ be the block projector that right-multiplies each layer's weight gradient by its $\mathsf{P}^{(l)}$. Under Assumption A2, the LANCE update $\boldsymbol{\theta}_{k+1} = \boldsymbol{\theta}_k - \eta \, \mathsf{Q} \, \nabla \mathcal{L}(\boldsymbol{\theta}_k)$ with $\eta \in (0, 1/L]$ satisfies $\mathcal{L}(\boldsymbol{\theta}_{k+1}) \leq \mathcal{L}(\boldsymbol{\theta}_k) - \frac{\eta}{2} \|\mathsf{Q} \nabla \mathcal{L}(\boldsymbol{\theta}_k)\|_2^2$, hence $\|\mathsf{Q} \nabla \mathcal{L}(\boldsymbol{\theta}_k)\|_2 \to 0$ and every limit point $\boldsymbol{\theta}_\star$ obeys $\mathsf{Q} \nabla \mathcal{L}(\boldsymbol{\theta}_\star) = \boldsymbol{0}$.*

*Proof.* By the Descent Lemma for an $L$-smooth function,

$$\mathcal{L}(\boldsymbol{\theta} - \eta \boldsymbol{g}) \leq \mathcal{L}(\boldsymbol{\theta}) - \eta \langle \nabla \mathcal{L}(\boldsymbol{\theta}), \boldsymbol{g} \rangle + \frac{L\eta^2}{2} \|\boldsymbol{g}\|_2^2.$$

Set $\boldsymbol{g} = \mathsf{Q} \, \nabla \mathcal{L}(\boldsymbol{\theta})$. Because $\mathsf{Q}$ is an orthogonal projector (blockwise right-multiplication by $\mathsf{P}^{(l)}$), $\langle \nabla \mathcal{L}, \mathsf{Q} \nabla \mathcal{L} \rangle = \|\mathsf{Q} \nabla \mathcal{L}\|_2^2$. Hence

$$\mathcal{L}(\boldsymbol{\theta}_{k+1}) \leq \mathcal{L}(\boldsymbol{\theta}_k) - \eta \|\mathsf{Q} \nabla \mathcal{L}(\boldsymbol{\theta}_k)\|_2^2 + \frac{L\eta^2}{2} \|\mathsf{Q} \nabla \mathcal{L}(\boldsymbol{\theta}_k)\|_2^2 \leq \mathcal{L}(\boldsymbol{\theta}_k) - \frac{\eta}{2} \|\mathsf{Q} \nabla \mathcal{L}(\boldsymbol{\theta}_k)\|_2^2$$

for $\eta \leq 1/L$. Summation gives $\sum_k \|Q\nabla\mathcal{L}(\boldsymbol{\theta}_k)\|_2^2 < \infty$ and thus $\|Q\nabla\mathcal{L}(\boldsymbol{\theta}_k)\|_2 \to 0$. Any limit point $\boldsymbol{\theta}_\star$ satisfies $Q\nabla\mathcal{L}(\boldsymbol{\theta}_\star) = \mathbf{0}$. $\square$

**Proposition 2.3** (Stationarity gap vs. truncation). *If along the iterates $\|(I - Q)\nabla\mathcal{L}(\boldsymbol{\theta})\|_2 \leq C\sqrt{1-\varepsilon}$ for some $C > 0$, then any limit point $\boldsymbol{\theta}_\star$ satisfies $\|\nabla\mathcal{L}(\boldsymbol{\theta}_\star)\|_2 \leq C\sqrt{1-\varepsilon}$.*

*Proof.* From Theorem 2.2, $\|Q\nabla\mathcal{L}(\boldsymbol{\theta}_k)\|_2 \to 0$. Then

$$\|\nabla\mathcal{L}(\boldsymbol{\theta}_k)\|_2 \ \leq \ \|Q\nabla\mathcal{L}(\boldsymbol{\theta}_k)\|_2 + \|(I - Q)\nabla\mathcal{L}(\boldsymbol{\theta}_k)\|_2 \ \to \ \leq \ C\sqrt{1-\varepsilon}.$$

Take a limit point $\boldsymbol{\theta}_\star$. $\square$

*Remark* B.1 (Why the leakage bound is reasonable). Right-multiplication by $\mathsf{P}^{(l)}$ removes only the gradient components aligned with the discarded input subspaces. One-shot HOSVD calibration bounds the relative energy of those components by $\sqrt{1-\varepsilon}$ in Frobenius norm, and layer-wise Lipschitz constants contract this via the chain rule to produce the stated bound.

## C COMPLEXITY ANALYSIS OF LANCE

Let $\mathbf{X}^{(l)} \in \mathbb{R}^{n_1 \times n_2 \times n_3}$, its compressed core $\mathbf{G}^{(l)} \in \mathbb{R}^{r_1 \times r_2 \times r_3}$, and the next-layer output $\mathbf{X}^{(l+1)} \in \mathbb{R}^{n_1 \times n_2 \times \hat{n}_3}$. Assume parameters $\boldsymbol{\theta}^{(l)} \in \mathbb{R}^{n_3 \times \hat{n}_3}$ operate on the last dimension. We estimate FLOPs and memory during fine-tuning and omit the one-shot HOSVD cost since it is performed offline.

The forward compression cost is $\sum_{i=1}^{3} n_i r_i \prod_{j \neq i} r_j$, the cost of projecting $\mathbf{X}^{(l)}$ with $\{\boldsymbol{U}_i^{(l)}\}_{i=1}^{3}$. Adding the layer's standard forward cost $\hat{n}_3 \prod_{i=1}^{3} n_i$ gives:

$$\mathrm{FLOPs}_{\mathrm{LANCE}}^{\mathrm{FW}} \ = \ \hat{n}_3 \prod_{i=1}^{3} n_i \ + \ n_1 r_1 n_2 n_3 \ + \ r_1 n_2 r_2 n_3 \ + \ r_1 r_2 n_3 r_3. \tag{7}$$

Similarly, the backward pass FLOPs with LANCE can be estimated as:

$$\mathrm{FLOPs}_{\mathrm{LANCE}}^{\mathrm{BW}} \ = \ \prod_{i=1}^{3} n_i r_1 \ + \ \prod_{i=1}^{3} r_i n_2 \ + \ r_1 n_2 r_3 n_3 \ + \ r_1 n_2 n_3 \hat{n}_3. \tag{8}$$

For full BP, the forward and backward FLOPs are:

$$\mathrm{FLOPs}_{\mathrm{BP}}^{\mathrm{FW}} \ = \ \hat{n}_3 \prod_{i=1}^{3} n_i, \quad \mathrm{FLOPs}_{\mathrm{BP}}^{\mathrm{BW}} \ = \ \hat{n}_3 \prod_{i=1}^{3} n_i. \tag{9}$$

Therefore, the overall improvement is

$$S_{\mathrm{FLOPs}} \ = \ \frac{\mathrm{FLOPs}_{\mathrm{BP}}^{\mathrm{FW}} + \mathrm{FLOPs}_{\mathrm{BP}}^{\mathrm{BW}}}{\mathrm{FLOPs}_{\mathrm{LANCE}}^{\mathrm{FW}} + \mathrm{FLOPs}_{\mathrm{LANCE}}^{\mathrm{BW}}}.$$

Since $r_i \ll n_i$, we obtain an improvement over BP on the order of $\sim 1.5\times$, consistent with Fig. 1c.

For memory usage, the improvement is measured as

$$S_{\mathrm{Mem}} \ = \ \frac{\mathrm{Mem}_{\mathrm{BP}}}{\mathrm{Mem}_{\mathrm{LANCE}}} \ = \ \frac{\prod_{i=1}^{3} n_i}{\prod_{i=1}^{3} r_i + \sum_{i=1}^{3} n_i r_i}.$$

Thus, LANCE reduces activation storage by a factor of $S_{\mathrm{Mem}}$ with negligible amortized overhead, since the decomposition is one-shot rather than repeated every iteration.

Table 8: Performance and efficiency comparison. Results are reported in terms of accuracy, memory usage, and Giga FLOPS (GFLOPs) for different batch sizes (BS).

| Mdl. | Method | # Layers | CIFAR10 (BS = 4) | | | CIFAR10 (BS = 8) | | | CIFAR100 (BS = 4) | | | CIFAR100 (BS = 8) | | |
|---|---|---|---|---|---|---|---|---|---|---|---|---|---|---|
| | | | Acc ↑ | MB ↓ | GFLOPS ↓ | Acc ↑ | MB ↓ | GFLOPS ↓ | Acc ↑ | MB ↓ | GFLOPS ↓ | Acc ↑ | MB ↓ | GFLOPS ↓ |
| MCUNet | BP | 2 | 77.17 | 0.220 | 0.025 | 76.87 | 0.439 | 0.049 | 54.24 | 0.220 | 0.025 | 53.96 | 0.439 | 0.049 |
| | | 4 | 83.93 | 0.403 | 0.039 | 83.68 | 0.806 | 0.079 | 60.99 | 0.403 | 0.039 | 59.27 | 0.806 | 0.079 |
| | LANCE | 2 | 69.72 | 0.007 | 0.014 | 76.32 | 0.011 | 0.028 | 45.70 | 0.008 | 0.014 | 51.27 | 0.010 | 0.028 |
| | | 4 | 74.23 | 0.018 | 0.023 | 77.89 | 0.020 | 0.045 | 49.23 | 0.015 | 0.023 | 53.36 | 0.019 | 0.045 |
| ResNet34 | BP | 2 | 94.08 | 0.766 | 1.850 | 93.92 | 1.531 | 3.699 | 77.85 | 0.766 | 1.850 | 77.99 | 1.531 | 3.699 |
| | | 4 | 94.96 | 1.531 | 3.699 | 95.10 | 3.062 | 7.399 | 79.02 | 1.531 | 3.699 | 79.10 | 3.062 | 7.399 |
| | LANCE | 2 | 91.75 | 0.059 | 1.018 | 92.29 | 0.093 | 2.022 | 73.36 | 0.061 | 1.021 | 74.51 | 0.101 | 2.036 |
| | | 4 | 92.05 | 0.168 | 2.114 | 93.20 | 0.232 | 4.124 | 73.09 | 0.156 | 2.096 | 76.17 | 0.252 | 4.161 |

Table 9: Performance and efficiency comparison. Results are reported in terms of accuracy, memory usage, and Giga FLOPS (GFLOPs) for different batch sizes (BS).

| Mdl. | Method | # Layers | CUB200 (BS = 4) | | | CUB200 (BS = 8) | | | PETS (BS = 4) | | | PETS (BS = 8) | | |
|---|---|---|---|---|---|---|---|---|---|---|---|---|---|---|
| | | | Acc ↑ | MB ↓ | GFLOPS ↓ | Acc ↑ | MB ↓ | GFLOPS ↓ | Acc ↑ | MB ↓ | GFLOPS ↓ | Acc ↑ | MB ↓ | GFLOPS ↓ |
| MCUNet | BP | 2 | 32.57 | 0.220 | 0.025 | 32.84 | 0.439 | 0.049 | 55.52 | 0.220 | 0.025 | 55.55 | 0.439 | 0.049 |
| | | 4 | 37.50 | 0.403 | 0.039 | 37.04 | 0.806 | 0.079 | 61.98 | 0.403 | 0.039 | 61.35 | 0.806 | 0.079 |
| | LANCE | 2 | 34.79 | 0.011 | 0.015 | 34.97 | 0.013 | 0.029 | 60.32 | 0.011 | 0.015 | 59.85 | 0.015 | 0.029 |
| | | 4 | 35.90 | 0.024 | 0.024 | 36.12 | 0.029 | 0.047 | 60.75 | 0.028 | 0.024 | 59.72 | 0.032 | 0.047 |
| ResNet34 | BP | 2 | 66.50 | 0.766 | 1.850 | 66.86 | 1.531 | 3.699 | 91.44 | 0.766 | 1.850 | 91.11 | 1.531 | 3.699 |
| | | 4 | 69.09 | 1.531 | 3.699 | 69.57 | 3.062 | 7.399 | 91.14 | 1.531 | 3.699 | 91.63 | 3.062 | 7.399 |
| | LANCE | 2 | 65.53 | 0.053 | 1.006 | 64.81 | 0.094 | 2.022 | 91.20 | 0.075 | 1.043 | 90.87 | 0.111 | 2.055 |
| | | 4 | 65.72 | 0.160 | 2.102 | 66.31 | 0.220 | 4.102 | 90.84 | 0.182 | 2.136 | 90.92 | 0.275 | 4.202 |

## D  ADDITIONAL GRADIENT-COMPRESSION ABLATIONS

For completeness, we extend the gradient fidelity analysis of Sec. 3.3 to three additional fine-tuning datasets: Flowers102, Oxford-IIIT Pets, and CIFAR-10. Using the same experimental setup as in the main paper, we vary the energy threshold $\varepsilon$ to induce different activation compression ratios and measure (i) the angle between full-BP gradients and LANCE-projected gradients, (ii) the resulting memory ratio, and (iii) the accuracy obtained under each setting. Appendix Fig. 5 shows results for MCUNet and ResNet34. The behavior is highly consistent with the CUB-200 and CIFAR-100 experiments: gradient alignment degrades smoothly as compression increases, yet even aggressive compression ($\sim$2 orders of magnitude memory reduction) preserves descent directions within $< 70°$. These findings demonstrate that the one-shot activation subspace generalizes well across models and datasets, further validating the stability and robustness of LANCE's low-rank representations.

## E  ROBUSTNESS OF LANCE UNDER SMALL-BATCH REGIMES

To evaluate LANCE under more restrictive memory settings, we additionally benchmarked batch sizes BS = 4 and BS = 8, which are more representative of MCU- and IoT-class devices. The results (Tables 8–9) show that LANCE maintains the same qualitative behavior observed with BS = 128: memory reductions of 1–2 orders of magnitude, negligible FLOPs overhead, and accuracy comparable to full BP. This indicates that LANCE's efficiency gains do not rely on large batch sizes and extend reliably to resource-constrained edge settings.

## F  LIMITATIONS AND FUTURE DIRECTION.

LANCE opens several promising directions for future work. First, while the method relies on activation subspaces obtained from a small calibration set, developing mechanisms to adapt or refine these subspaces under strong domain shift or evolving representations would broaden its applicability—including extending the approach to earlier stages of *pre-training*, where representation drift is more pronounced. In continual learning, relaxing the strict orthogonality constraint could enable

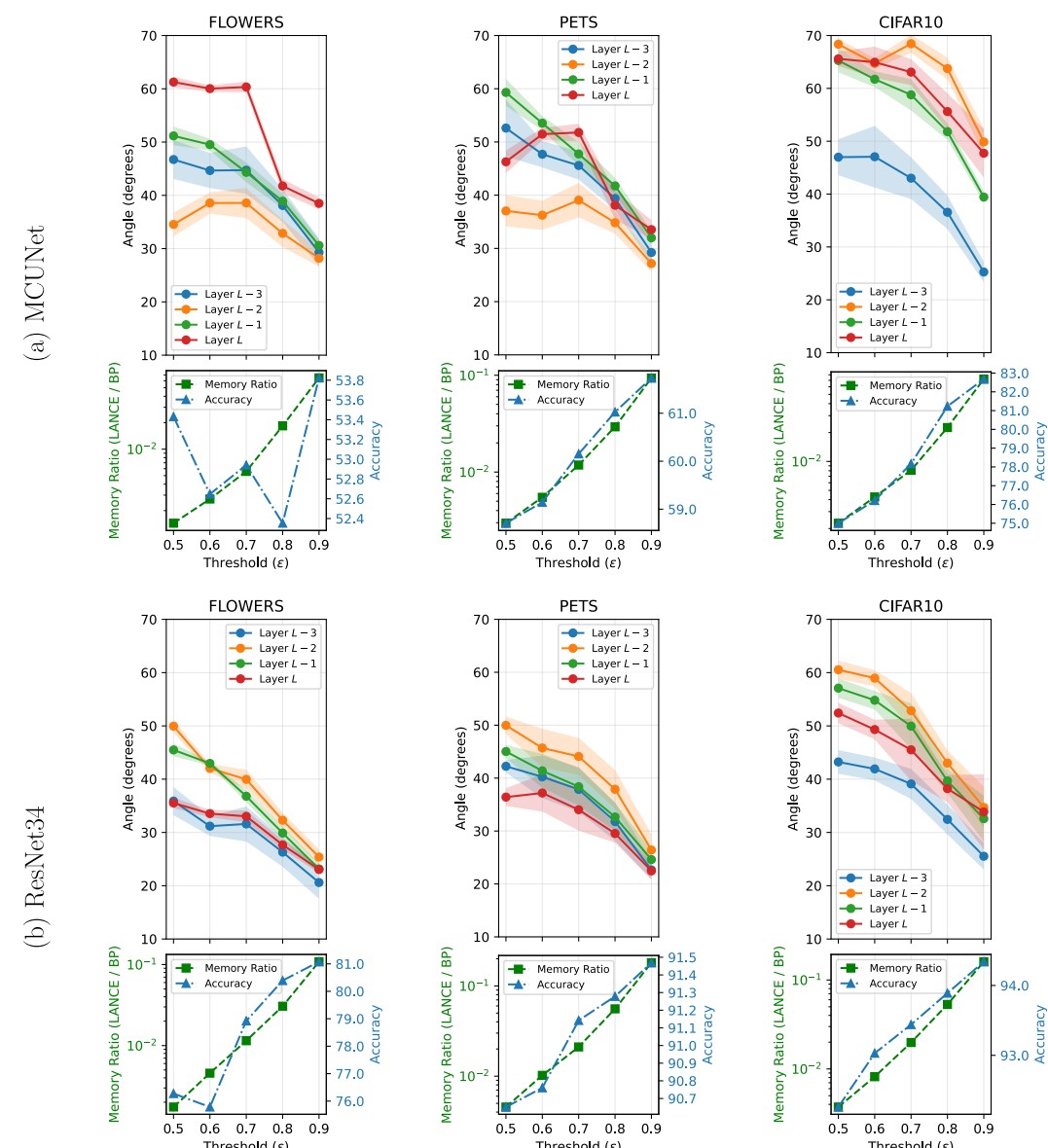

Figure 5: Additional gradient–compression ablations on Flowers, Pets, and CIFAR-10. We repeat the analysis of Fig. 4 on three additional fine-tuning datasets using MCUNet and ResNet34. For each dataset, we report the mean and standard deviation of the angle between full-BP gradients and LANCE-projected gradients over the final 10 epochs of training, together with the corresponding memory ratio (LANCE/BP) and accuracy across different energy thresholds $\varepsilon$. Across all datasets and architectures, gradient alignment degrades smoothly as $\varepsilon$ decreases, while extreme compression (up to two orders of magnitude in activation memory) continues to yield descent directions within $\sim 70°$ of full BP. These trends closely mirror those observed in the main experiments, reinforcing the robustness and consistency of LANCE's low-rank subspace across diverse data distributions.

positive forward transfer between related tasks while retaining stability. Additionally, augmenting the fixed low-rank subspace with selective or lightweight updates may improve adaptability on tasks requiring larger representational changes. Finally, although LANCE is compatible with MCU-class devices, integrating it with device-specific compilation and memory-scheduling pipelines presents an exciting opportunity to realize its full hardware potential.

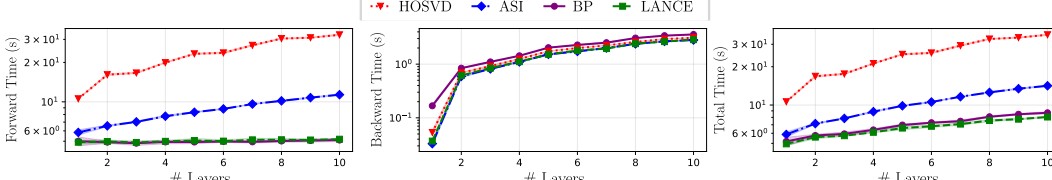

Figure 6: End-to-end fine-tuning latency on a Raspberry Pi 3 Model B+ (Broadcom BCM2837B0, quad-core Cortex-A53) when training MCUNet for 5 iterations on CIFAR-10 using batch size of 128. Each curve shows the mean over five independent trials. LANCE achieves the lowest forward, backward, and total training time across all layer depths. Unlike iterative low-rank methods such as ASI and HOSVD, which incur substantial forward-time overhead due to repeated decompositions, LANCE remains close to full BP while consistently outperforming all baselines, confirming its practical efficiency on real edge hardware.

## G EXPERIMENTAL VALIDATION ON RESOURCE-CONSTRAINED EDGE DEVICES

We evaluated LANCE on real edge hardware to validate its practical runtime efficiency. All methods were deployed on a Raspberry Pi 3 Model B+ (Broadcom BCM2837B0, quad-core ARM Cortex-A53), where we fine-tuned an MCUNet model for five iterations on CIFAR-10 using batch size of 128. Each configuration was run for five independent trials, and we report mean end-to-end latency across 1–10 trainable layers. As shown in Fig. 6, LANCE consistently achieves the lowest forward, backward, and total training time. Relative to LANCE, BP is 3–9% slower, ASI is 19–76% slower, and iterative HOSVD is $2.1\times$–$4.4\times$ slower, demonstrating that one-shot subspace calibration eliminates the overhead of repeated decompositions and yields tangible speedups on resource-constrained devices.

## H USE OF LARGE LANGUAGE MODELS (LLMS)

In accordance with the submission guidelines, we disclose the use of large language models (LLMs) during the preparation of this manuscript. LLMs were employed solely as a general-purpose writing assistant to improve clarity, grammar, and style. They were not used for research ideation, algorithm design, data analysis, or experiment implementation.

## I CODE AVAILABILITY

The implementation of LANCE, together with scripts to reproduce all experiments, will be released in a public repository after the review process. This will include training and evaluation code, dataset preprocessing pipelines, and experiment configuration files to ensure full reproducibility.

