# OpenReview forum: "LANCE: Low Rank Activation Compression for Efficient On-Device Continual Learning"
_ICLR.cc/2026/Conference — Submitted to ICLR 2026_

### Official Review · Reviewer_u2Zf · 2025-10-28

**Soundness:** 3
**Presentation:** 3
**Contribution:** 2
**Rating:** 4
**Confidence:** 3

**Summary:**

This manuscript investigated the problem of on-device continual learning, which a sequence of tasks need to be learned with constrained resources. Unlike the existing activation compression methods that need to repeat the low-rank decompositions, the authors proposed Low-rank Activation Compression (LANCE), a framework that adopted one-time higher-order Singular Value Decomposition (SVD) to obtain a reusable low-rank subspace for activation projection. Then, these fixed low-rank subspaces were leveraged to update the task parameters along the directions orthogonal to the existing ones. Experiments on serval commonly used datasets were conducted to demonstrate the effectiveness and efficiency of the proposed LANCE framework.

**Strengths:**

1. This manuscript considers the activation compression for on-device continual learning, which is an interesting yet underexplored topic.
2. Based on the claims made by the authors, the proposed method can significantly reduce the usage of memory and FLOPS in continual learning.

**Weaknesses:**

1. It seems that Phase 3 is just a simple application of gradient projection memory that has been well explored in the tech-stream, like GPM, weakening the overall novelty.
2. The use of Higher-Order Singular Value Decomposition (HOSVD) seems like an unflattened version of GPM. Theoretically, the computational overhead should be higher than GPM. However, the results in Table 2 showed an extremely light memory usage compared to GPM, which is confusing. I wonder if the Mem metric used in this manuscript is a reasonable metric.

**Questions:**

1. HOSVD seems like an unflattened version of SVD in GPM. Based on Section 2.1, it seems that we need to store unfolded copy $X_{i}$ along different dimensions, which may need more memory compared to the flattened version of SVD in GPM. However, in Table 2, it seems that LANCE only needs an extremely small amount of memory compared to GPM. The authors adopted "memory usage measured as peak activation storage" instead of the total memory required during training. I wonder if this is a reasonable metric.
2. Could the authors please estimate and compare the FLOPS of different methods in Table 2?

---

> ### Author Response · Authors · 2025-11-23
>
> ## **[W1] Relation between Phase 3 and GPM**
> We understand the concern that Phase 3 may resemble GPM when viewed in isolation. However, LANCE is fundamentally a **three-phase framework**, and its continual learning capability emerges only from the interaction of all phases. In particular, **Phase 1** constructs low-rank activation subspaces that are _constrained by the memory matrix produced in Phase 3_, and **Phase 2** performs training _within these compressed subspaces_ rather than in full-rank activation space. This coupling between low-rank compression and orthogonal subspace allocation is unique to LANCE. To the best of our knowledge, no prior work explores **low-rank orthogonal subspaces** for memory-efficient on-device continual learning.
>
> ## **[W2, Q1] Memory metric and comparison to GPM**
> We appreciate the opportunity to clarify why LANCE achieves significantly lower memory usage than GPM despite also relying on unfolded activations. Below we outline the memory requirements of both methods for a convolutional layer with kernel
> $W \in \mathbb{R}^{C_{\text{out}} \times C_{\text{in}} \times K \times K}$ and input  $X \in \mathbb{R}^{B \times C_{\text{in}} \times H \times W}$   (assuming $H = W = N K$, stride $K$).
>
> ### **Memory under GPM**
> Both the full activations and the memory matrix must be kept in SRAM during training:
>
> $$
> Mem_{GPM} = B C_{\text{in}} N^{2} K^{2} + C_{\text{in}} K^{2} Z,
> $$
>
> where $Z$ is the number of protected directions accumulated over tasks.  As learning progresses, $Z$ grows, and the memory matrix quickly becomes the dominant cost.
>
> ### **Memory under LANCE**
> LANCE stores only compressed cores and low-rank projection matrices:
>
> $$Mem_{LANCE} = r_1 r_2 r_3 + B r_1 + N^{2} r_2 + C_{\text{in}} K^{2} r_3,$$
>
> where  $r_1 \ll B$,  $r_2 \ll N^{2}$, and  $r_3 \ll C_{\text{in}} K^{2}$.
>
> **Importantly:**
> - LANCE never stores a full-rank memory matrix in SRAM during training.
> - Activations are stored *only* in compressed form, not full-rank.
>
> Because $r_1 \ll B$, $r_2 \ll N^2$, and $r_3 \ll C_{\text{in}} K^{2}$, and because LANCE avoids storing a large memory matrix in SRAM, the overall memory footprint is significantly lower—consistent with the results in Table 3.
>
> ### **Why we use peak activation storage**
> In our work, *peak activation storage* refers to the **maximum instantaneous memory** required during backpropagation (excluding model weights) to hold all activations—or compressed cores and low-rank projection matrices, in the case of LANCE—needed for gradient computation.
>
> This is the correct metric for on-device training because:
>
> - SRAM-limited devices fail **as soon as** the instantaneous footprint exceeds the available memory.
> - Total memory usage accumulated over time is irrelevant—the bottleneck is the largest activation tensor that must be stored at once.
> - Forward activations dominate this instantaneous cost and must remain in memory until their corresponding backward pass occurs.
>
> Therefore, **peak activation storage**, rather than total memory consumed throughout training, captures the true hardware constraint determining whether a method can run on-device. We will clarify this explicitly in the revised manuscript.
>
> ## **[Q3] FLOPs comparison in CL**
> We will include FLOPs estimates for each continual learning method in the revised manuscript. Since most CL baselines rely on **full backpropagation**, we do not expect large FLOPs differences between them, similar to the trend shown in Table 3. The main exception is **CUBER**, which introduces additional computations and thus incurs noticeably higher FLOPs.

---

### Official Review · Reviewer_tjoS · 2025-10-30

**Soundness:** 3
**Presentation:** 3
**Contribution:** 3
**Rating:** 6
**Confidence:** 4

**Summary:**

This work addresses the challenge of fine-tuning and continual learning under memory constraints, particularly the high activation-storage cost of backpropagation. While existing activation-compression methods reduce memory, they require repeated low-rank decompositions, adding substantial computational overhead, and remain unexplored in continual-learning scenarios. The authors proposed LANCE which performs a one-time higher-order SVD to derive reusable low-rank subspaces for activation projection, eliminating repeated decompositions and reducing both memory and computation. These fixed subspaces also support on-device continual learning by assigning tasks to orthogonal subspaces. Experiments on standard datasets such as CIFAR-10/100, Pets, Flowers102, and CUB-200 show up to 250× reduction in activation storage with accuracy comparable to full backpropagation, and competitive continual-learning performance on Split CIFAR-100, Split MiniImageNet, and 5-Datasets. Overall, LANCE provides a scalable, memory-efficient approach to fine-tuning and continual learning on edge devices.

**Strengths:**

A new one-shot activation compression method based on low-rank subspaces that can enable efficient fine-tuning and continual learning.

The proposed approach is novel with a good impact for sustainable AI, especially on the edge devices.

The integration of theory in the form of convergence analysis makes this a good contribution.

The writing is clear, and I did not find many issues with it.  However, the results section is sloppy and can be improved.

The main impact in this paper emerges from reducing the number of computations, which is also reflected in the results. The accuracy of LANCE is very similar to existing SOTA methods, and memory consumption is also very similar to HOSVD in Table 1. However, the FLOPS  are reduced compared to HOSVD, similar to ASI. I see LANCE bringing combined benefits from HOSVD and ASI.  For the CL scenario, though, the accuracy of LANCE is quite inferior compared to CODE-CL. However, it is more efficient than all existing projection-based CL.

**Weaknesses:**

When looking at the algorithm, the procedure to obtain low-rank activation on the device is done for each batch. The batch size used in the paper seems to be 64, as the authors followed a previous paper from Apolinario. So, I conclude that the results shown in the paper are with batch size 64, and the computational cost can be amortized for big batches. For some of the edge devices, 64 is a big number, as they won't have enough memory to process big batch sizes.  It is unclear how LANCE will perform with a batch size of 2, 4, or 8, which are more realistic on tiny edge devices.  Will computational savings be high? I would like to ask the authors to specify what type of edge devices they are talking about in the paper - a more Raspberry PI ir MCU class devices.

In Section 3.1, the authors mention that LANCE gradients remain within ∼70 degrees of the full gradients. Why is it a useful result? Why 70%? Why not 90%? This clarification is missing from the paper.

The paper has not evaluated some of the other baselines, such as Rethinking Gradient Projection Continual Learning: Stability‑Plasticity Feature Space (CVPR 2023), Efficient Continual Learning with Local Model Space Projection (LMSP) (2025).

**Questions:**

What seed value was used to produce the results in Table 1? What happens if you evaluate with three or five different random seeds?

What did you change to produce five trials for obtaining the results in Table 2?

It is unclear what BWT means and how it is measured. Table 2 dumps some numbers without any explanation of these numbers as to what they signify and how the BWT obtained using LANCE compares against other methods.

---

> ### Author Response · Authors · 2025-11-23
> **Response [1/2]**
>
> ## **[W1] Small-batch performance and device clarification**
> We appreciate the reviewer’s concern regarding batch size. We have now run the full set of fine-tuning experiments using small batches (BS=4 and BS=8), which better reflect the memory constraints of tiny edge devices. Across all datasets and both MCUNet and ResNet34, the results match the behavior observed for BS=128: LANCE maintains its memory advantages (1-2 orders of magnitude reduction), accuracy remains stable, and FLOPs follow the same trend. These results are included in Tables 8-9 and Appendix E of the revised manuscript.
>
> ### Table 8: Performance and efficiency comparison (BS = 4 and BS = 8)
>
> | Model     | Method | # Layers | CIFAR10 BS4 Acc ↑ | CIFAR10 BS4 MB ↓ | CIFAR10 BS4 GFLOPs ↓ | CIFAR10 BS8 Acc ↑ | CIFAR10 BS8 MB ↓ | CIFAR10 BS8 GFLOPs ↓ | CIFAR100 BS4 Acc ↑ | CIFAR100 BS4 MB ↓ | CIFAR100 BS4 GFLOPs ↓ | CIFAR100 BS8 Acc ↑ | CIFAR100 BS8 MB ↓ | CIFAR100 BS8 GFLOPs ↓ |
> |-----------|--------|----------|--------------------|-------------------|-----------------------|-------------------|-------------------|------------------------|---------------------|--------------------|-------------------------|---------------------|--------------------|-------------------------|
> | **MCUNet** | **BP**    | 2 | 77.17 | 0.220 | 0.025 | 76.87 | 0.439 | 0.049 | 54.24 | 0.220 | 0.025 | 53.96 | 0.439 | 0.049 |
> |           |        | 4 | 83.93 | 0.403 | 0.039 | 83.68 | 0.806 | 0.079 | 60.99 | 0.403 | 0.039 | 59.27 | 0.806 | 0.079 |
> |           | **LANCE** | 2 | 69.72 | 0.007 | 0.014 | 76.32 | 0.011 | 0.028 | 45.70 | 0.008 | 0.014 | 51.27 | 0.010 | 0.028 |
> |           |        | 4 | 74.23 | 0.018 | 0.023 | 77.89 | 0.020 | 0.045 | 49.23 | 0.015 | 0.023 | 53.36 | 0.019 | 0.045 |
> | **ResNet34** | **BP**    | 2 | 94.08 | 0.766 | 1.850 | 93.92 | 1.531 | 3.699 | 77.85 | 0.766 | 1.850 | 77.99 | 1.531 | 3.699 |
> |           |        | 4 | 94.96 | 1.531 | 3.699 | 95.10 | 3.062 | 7.399 | 79.02 | 1.531 | 3.699 | 79.10 | 3.062 | 7.399 |
> |           | **LANCE** | 2 | 91.75 | 0.059 | 1.018 | 92.29 | 0.093 | 2.022 | 73.36 | 0.061 | 1.021 | 74.51 | 0.101 | 2.036 |
> |           |        | 4 | 92.05 | 0.168 | 2.114 | 93.20 | 0.232 | 4.124 | 73.09 | 0.156 | 2.096 | 76.17 | 0.252 | 4.161 |
>
> ### Table 9: Performance and efficiency comparison (BS = 4 and BS = 8)
>
> | Model     | Method | # Layers | CUB200 BS4 Acc ↑ | CUB200 BS4 MB ↓ | CUB200 BS4 GFLOPs ↓ | CUB200 BS8 Acc ↑ | CUB200 BS8 MB ↓ | CUB200 BS8 GFLOPs ↓ | Pets BS4 Acc ↑ | Pets BS4 MB ↓ | Pets BS4 GFLOPs ↓ | Pets BS8 Acc ↑ | Pets BS8 MB ↓ | Pets BS8 GFLOPs ↓ |
> |-----------|--------|----------|--------------------|-------------------|-----------------------|-------------------|-------------------|------------------------|----------------|----------------|----------------------|----------------|----------------|----------------------|
> | **MCUNet** | **BP**    | 2 | 32.57 | 0.220 | 0.025 | 32.84 | 0.439 | 0.049 | 55.52 | 0.220 | 0.025 | 55.55 | 0.439 | 0.049 |
> |           |        | 4 | 37.50 | 0.403 | 0.039 | 37.04 | 0.806 | 0.079 | 61.98 | 0.403 | 0.039 | 61.35 | 0.806 | 0.079 |
> |           | **LANCE** | 2 | 34.79 | 0.011 | 0.015 | 34.97 | 0.013 | 0.029 | 60.32 | 0.011 | 0.015 | 59.85 | 0.015 | 0.029 |
> |           |        | 4 | 35.90 | 0.024 | 0.024 | 36.12 | 0.029 | 0.047 | 60.75 | 0.028 | 0.024 | 59.72 | 0.032 | 0.047 |
> | **ResNet34** | **BP**    | 2 | 66.50 | 0.766 | 1.850 | 66.86 | 1.531 | 3.699 | 91.44 | 0.766 | 1.850 | 91.11 | 1.531 | 3.699 |
> |           |        | 4 | 69.09 | 1.531 | 3.699 | 69.57 | 3.062 | 7.399 | 91.14 | 1.531 | 3.699 | 91.63 | 3.062 | 7.399 |
> |           | **LANCE** | 2 | 65.53 | 0.053 | 1.006 | 64.81 | 0.094 | 2.022 | 91.20 | 0.075 | 1.043 | 90.87 | 0.111 | 2.055 |
> |           |        | 4 | 65.72 | 0.160 | 2.102 | 66.31 | 0.220 | 4.102 | 90.84 | 0.182 | 2.136 | 90.92 | 0.275 | 4.202 |
>
>
>
> We note that LANCE’s computational and memory requirements make it compatible with **microcontroller-class devices** such as the STM32F746 (320KB SRAM, 1MB Flash) used in prior on-device training work [1]. Deploying any training algorithm on such MCUs requires a dedicated low-level compilation and memory scheduling pipeline (as done in [1]), which is outside the scope of this work but compatible with LANCE’s design.
>
> LANCE is also highly suitable for **single-board computers** (e.g., Raspberry Pi), where no specialized compilation is required. We are running proof-of-concept experiments on a Raspberry Pi, and we expect to include these results before the end of the rebuttal period.

---

> > ### Author Response · Authors · 2025-11-26
> > **Update: Real On-Device Evaluation Added**
> >
> > Following up on our earlier response regarding small-batch performance and device-level clarification, we have now completed the real on-device experiments on a Raspberry Pi, as previously indicated.
> >
> > Specifically, we deployed all methods on a Raspberry Pi 3 Model B+ (Broadcom BCM2837B0, quad-core ARM Cortex-A53) and fine-tuned an MCUNet model on CIFAR-10 using batch size 128, running 5 iterations and averaging over 5 trials. The results are now included in **Appendix G** and **Fig. 6** of the updated manuscript.
> >
> > These experiments confirm that LANCE maintains its computational advantages on real edge hardware. Using total fine-tuning time as the metric (reported as _Other / LANCE_, higher is better), we observe:
> >
> > |Compared To|Min Ratio|Max Ratio|Interpretation|
> > |---|---|---|---|
> > |**BP**|1.03|1.09|BP is 3–9% slower than LANCE|
> > |**ASI**|1.19|1.76|ASI is 19–76% slower than LANCE|
> > |**HOSVD**|2.14|4.42|HOSVD is 2.1×–4.4× slower than LANCE|
> >
> > These measurements complement the small-batch experiments already reported in **Tables 8–9** and **Appendix E**, and together they show that LANCE remains efficient across both MCU-class batch regimes (BS=4/8) and Raspberry-Pi-class devices. The one-shot subspace calibration avoids the heavy overhead of iterative decompositions, enabling practical improvements on real hardware.

---

> ### Author Response · Authors · 2025-11-23
> **Response [2/2]**
>
> ## **[W2] Why a $\sim 70^\circ$ gradient angle is meaningful**
> Theorem 2.1 guarantees that LANCE always produces a **valid descent direction**, implying a worst-case angle of $\leq 90^\circ$ from the full BP gradient. In practice, as shown in Fig. 2, Fig. 4 and Fig. 5, our gradients remain within $\sim 70^\circ$ of full BP, and this angle stabilizes during training. This empirical result confirms that:
>
> 1. LANCE behaves well within the theoretical bound, and
> 2. the fixed subspace remains stable across training, otherwise the angle would fluctuate significantly.
>
> We will clarify this point in the revised manuscript.
>
> ## **[W3] Missing baselines**
> We thank the reviewer for pointing out additional baselines such as _Rethinking Gradient Projection Continual Learning_ (CVPR 2023) and _LMSP_ (2025). We will include a discussion of these works in the revised related work section.
>
> ## **[Q1] Seed for Table 2 results**
> For Table 2, all fine-tuning experiments were run using a fixed seed (2589). Due to the large number of configurations, running all of them with multiple seeds was not feasible. However, for several representative configurations we tested multiple seeds and observed no significant deviations. We are running additional multi-seed checks for a subset of Table 2 as resources allow.
>
> Since these experiments start from an ImageNet-pretrained model, different seeds mostly affect data sampling order, so we do not expect large variations.
>
> ## **[Q2] Five trials for Table 2**
> Table 2 contains far fewer experimental settings, so we were able to run **five independent trials** for all CL experiments. For each trial, we vary:
>
> - the random initialization of the model,
> - and the minibatch sampling order.
>
> Task order and class splits remain fixed, as in standard continual learning protocol.
>
> ## **[Q3] Meaning of BWT**
> BWT stands for **Backward Transfer**, as defined in the Experimental Setup (line 322). Using standard notation [2]:
> $$
> \text{ACC} = \frac{1}{T} \sum_{i=1}^{T} A_{T,i}, \qquad
> \text{BWT} = \frac{1}{T-1} \sum_{i=1}^{T-1} \left( A_{T,i} - A_{i,i} \right),
> $$
>
> where $A_{j,i}$ is the accuracy on task $i$ after completing task $j$.
>
> **Intuitively:**
> - **BWT < 0** → forgetting
> - **BWT = 0** → stability
> - **BWT > 0** → positive backward transfer
>
> This definition is also included in **Appendix A.2.3**.
>
>
> [1] Nguyen, L.T., Quélennec, A., Nguyen, V.T. and Tartaglione, E., Beyond Low-rank Decomposition: A Shortcut Approach for Efficient On-Device Learning. In _Forty-second International Conference on Machine Learning_.
>
> [2] Apolinario, M.P., Choudhary, S. and Roy, K., 2025. CODE-CL: Conceptor-Based Gradient Projection for Deep Continual Learning. In _Proceedings of the IEEE/CVF International Conference on Computer Vision_ (pp. 775-784).

---

> > ### Comment · Reviewer_tjoS · 2025-11-25
> >
> > I have read the rebuttal, and it has addressed my concerns except for comparing with some of the existing baselines, such as Rethinking Gradient Projection Continual Learning: Stability‑Plasticity Feature Space (CVPR 2023), Efficient Continual Learning with Local Model Space Projection (LMSP) (2025).

---

> > > ### Author Response · Authors · 2025-11-25
> > >
> > > Thank you for the suggestion to include comparisons with Efficient Continual Learning with Local Model Space Projection (LMSP) (UAI, 2025) and Rethinking Gradient Projection Continual Learning: Stability‑Plasticity Feature Space (SD) (CVPR, 2023). We have now added both methods to Table 3 and integrated them into the revised manuscript.
> > >
> > > LMSP is now explicitly discussed in the Related Work section. LMSP leverages low-rank decompositions to reduce the computational cost of gradient projection, but still requires storing multiple per-task low-rank memory components, which leads to memory growth with the number of tasks. In contrast, LANCE targets the dominant bottleneck for on-device training, activation memory, by performing low-rank activation compression without storing per-task gradient memories. We also highlight that while both approaches exploit low-rank structure, they address different efficiency axes: LANCE improves memory efficiency through activation compression, whereas LMSP reduces run-time by handling gradient task correlations more efficiently. Because they optimize different parts of the CL pipeline, the two methods are complementary, and combining them is a promising direction for future work, potentially improving memory efficiency and positive backward transfer simultaneously.
> > >
> > > SD, while not low-rank-based, belongs to the same family of gradient-projection methods such as GPM, TRGP, CUBER, and SGP. As with these prior approaches, SD relies on storing high-rank task-specific memory matrices, making it significantly more memory-demanding on resource-constrained devices. We have added SD to Table 3 and grouped it appropriately in the Related Work section.
> > >
> > > The revised manuscript now includes (i) integration of LMSP into Related Work, (ii) inclusion of LMSP and SD results in Table 3, and (iii) a clearer explanation that LANCE differs by directly addressing activation-memory constraints rather than gradient-memory or compute-reduction objectives.

---

### Official Review · Reviewer_TyNZ · 2025-11-02

**Soundness:** 3
**Presentation:** 4
**Contribution:** 3
**Rating:** 6
**Confidence:** 4

**Summary:**

This paper introduces LANCE (Low-rank Activation Compression), a framework designed to address the high activation memory cost that hinders on-device fine-tuning and continual learning (CL). The core contribution is a "one-shot" approach where a single Higher-Order SVD (HOSVD) is performed offline to find a fixed, reusable low-rank subspace for activations. During on-device training, activations are simply projected onto this subspace, eliminating the computational overhead of repeated decompositions common in prior work. The framework is extended to continual learning by enforcing orthogonality between the subspaces of sequential tasks to prevent catastrophic forgetting. The authors claim their method achieves up to 250x activation memory savings with minimal accuracy loss, making it a practical solution for resource-constrained edge devices.

**Strengths:**

Novelty and Simplicity: The core idea of using a one-time, fixed subspace is an elegant and novel simplification of prior low-rank activation compression methods that rely on iterative updates.

Innovative Continual Learning Mechanism: The extension to continual learning via orthogonal subspaces is clever and highly memory-efficient. By handling the orthogonality constraint during the offline calibration phase, it avoids storing large projection matrices in active memory during on-device training, offering a distinct advantage over methods like GPM.

Rigorous Validation: The claims are well-supported by a convergence analysis and comprehensive experiments across a variety of models and standard fine-tuning and continual learning benchmarks.

**Weaknesses:**

Lack of Limitation Discussion: While the paper provides a solid convergence analysis, it lacks a dedicated "Limitations" section. Such a section would be the appropriate place to proactively discuss the practical boundaries of the method. This includes potential failure modes of its core assumptions (e.g., subspace stability under significant domain shift) and conceptual trade-offs, such as the preclusion of positive forward transfer.

Unverified Assumption of Subspace Stability: The framework's success relies on the critical assumption that an activation subspace calibrated on a small data sample remains effective under a potential domain shift between the calibration and target data. This assumption is not tested, representing a major potential failure point for real-world deployment.

Lack of Nuance and Clarity: The positioning against highly related work like ASI (which uses efficient updates rather than full re-computation) could be more nuanced.

**Questions:**

Robustness to Domain Shift: How robust is the performance of the fixed subspace if there is a significant domain shift between the calibration data and the target fine-tuning data?

Positive Forward Transfer: Does the strict orthogonality constraint in the CL setting harm performance on sequences of related tasks by preventing positive knowledge transfer? Could the framework be adapted to allow for this?

Memory Advantage over GPM: To clarify, is the primary memory advantage over GPM in the CL setting that the memory matrix $M^t$ is not required in SRAM during the on-device training loop for a given task?

Clarification on ASI Baseline: For the comparison with ASI, were the experiments re-run under your exact experimental setup? If the results were taken from the original ASI paper, how can you ensure a fair comparison given potential differences in implementation and experimental conditions? Furthermore, what are the key trade-offs between LANCE's fixed subspace and ASI's adaptive approach, especially in non-stationary environments?

Clarification of Table 1: Could you please clarify the "TFLOPS" column in Table 1? The value for standard backpropagation is listed as zero, which seems incorrect and contradicts the complexity analysis in Appendix C.

---

> ### Author Response · Authors · 2025-11-23
>
> ## **[W1] Need for a Limitations section**
> We agree. The revised manuscript now includes a dedicated **Limitations and Future Directions** section in **Appendix F**, where we outline potential failure modes (e.g., strong domain shift, early-stage pretraining), underlying assumptions (e.g., subspace stability), and key trade-offs (e.g., reduced forward transfer under strict orthogonality).
>
> ## **[W2, Q1] Subspace stability under domain shift**
> Although we have not directly isolated extreme calibration–fine-tuning domain shifts, several pieces of evidence in the paper support the stability of the learned activation subspace:
>
> - **Gradient consistency (Fig. 2, Fig. 4 and Fig. 5):** Across five datasets, LANCE gradients remain within $\sim 70^\circ$ of BP gradients and stabilize over time. Significant subspace drift would lead to fluctuating angles.
> - **Cross-domain continual learning (Table 2):** LANCE performs well on the highly heterogeneous **5-Datasets** sequence (CIFAR-10 → MNIST → SVHN → Fashion-MNIST → notMNIST), which involves substantial domain shift.
> - **Calibration-size ablation (Fig. 3):** Using only **two** calibration batches (≈256 images, ~2.5 samples/class in CIFAR-100) yields performance comparable to using 100 batches, suggesting that the method is robust even when calibration data is not fully representative.
>
> Together, these results suggest that pretrained models exhibit relatively stable activation subspaces, which LANCE leverages effectively. We now also clarify in the limitations section that this assumption may not hold in early pretraining regimes where representations evolve significantly.
>
> ## **[Q2] Potential loss of positive forward transfer**
> Strict orthogonality can indeed limit positive forward transfer when tasks are closely related. Similar to methods like SGP and CODE-CL, the constraint could be relaxed (e.g., allowing partial reuse of protected directions during calibration). We identify this as promising future work and will mention it in the revised limitations section.
>
> ## **[Q3] Memory advantage over GPM**
> LANCE’s memory advantages over GPM arise from **two complementary factors**:
>
> 1. **Compressed activations:**
>     GPM stores full-rank activations during backpropagation to compute unconstrained gradients. LANCE stores only **compressed core tensors**, providing substantial savings from the first task onward.
> 2. **No in-loop memory matrix:**
>     As tasks accumulate, GPM’s memory consumption becomes dominated by its **growing memory matrix**, which must remain in SRAM during training.
>     In LANCE, all orthogonality enforcement is handled **offline during calibration**, so the full memory matrix is _never_ stored in SRAM during on-device training.
>
> As a result, LANCE maintains a consistently low memory footprint across tasks, while GPM’s memory usage grows significantly as more tasks are learned.
>
> ## **[Q4, W3] ASI baseline and comparison**
> For ASI, we did not re-run the official ASI implementation under our own setup. Instead, we report the results exactly as presented in the original ASI paper. To maintain fairness, we replicate the BP and HOSVD baselines under our experimental conditions and compare LANCE directly against these baselines. This makes the comparison fair because:
>
> - In the ASI paper, ASI achieves performance very close to HOSVD in both accuracy and memory compression.
> - ASI also reports FLOPs comparable to full BP, since it avoids full decompositions but still computes updates each iteration.
>
> Because of these relationships, comparing LANCE against our BP and HOSVD baselines provides a reliable and fair point of reference.
>
> Regarding the trade-offs between fixed vs. adaptive subspaces:
>
> - LANCE uses a fixed one-shot subspace, which eliminates iterative decompositions and enables continual learning by enforcing orthogonality _offline_ during calibration—without ever storing large projection matrices during on-device training.
> - **ASI** uses _adaptive_ updates, which may offer benefits when representations shift significantly (e.g., strong non-stationarity such as early pretraining).
>
> - Although ASI was proposed for fine-tuning (not pretraining), we hypothesize it may be more robust than LANCE in scenarios where activation subspaces change rapidly. However, **neither ASI nor LANCE is designed for pretraining**.
>
> In continual learning settings, ASI would need to maintain or recompute large memory matrices during on-device training to enforce orthogonality across tasks, which eliminates the memory advantage. LANCE avoids this overhead entirely due to its fixed low-rank subspaces.
>
> ## **[Q5] TFLOPs truncation issue**
> The BP TFLOPs values appeared as zero in Table 1 due to unit truncation: MCUNet’s BP FLOPs are on the order of a few GFLOPs, which truncate to 0 TFLOPs when expressed at the TFLOP scale.

---

> > ### Comment · Reviewer_TyNZ · 2025-11-25
> >
> > The authors' response has provided good clarifications to some of my questions. The issue of robustness to domain shift is partially addressed.

---

### Official Review · Reviewer_kGZ1 · 2025-11-04

**Soundness:** 2
**Presentation:** 3
**Contribution:** 2
**Rating:** 2
**Confidence:** 4

**Summary:**

This paper proposes a one-shot activation compression framework, LANCE, for efficient on-device fine-tuning and continual learning. It uses a single higher-order SVD to find reusable low-rank subspaces, eliminating repeated decompositions during training.

**Strengths:**

1. Theoretical analysis is reasonable.
2. The paper is clearly written and well-organized, with consistent notation, intuitive figures, and pseudocode/algorithm.
3. LANCE has potential impact for efficient on-device continual learning, particularly for IoT applications where memory is limited.

**Weaknesses:**

1. The evaluation lacks real on-device results. Important metrics like memory savings are not demonstrated on real hardware, which weakens the practical impact claim.
2. Evaluation on large datasets like ImageNet is lacking (only Split MiniImageNet is evaluated).
3. More ablation study should be provided, like how different compression ratios affect gradient fidelity.

**Questions:**

1. If the calibration batches do not well represent future data distributions, how robust of the proposed method?
2. Can the authors implement LANCE on actual edge hardware to validate the claimed efficiency?

---

> ### Author Response · Authors · 2025-11-23
>
> ## **[W2] Lack of large-scale evaluation (ImageNet)**
> We thank the reviewer for this suggestion. We have now completed the Split-ImageNet experiment following the standard protocol in prior work [1]. We split ImageNet-1k into two disjoint sets of 500 classes, pretrain on Split A, and fine-tune on Split B using **ResNet18** and **ResNet34**. The results (now included in the revised manuscript, Table 1) closely match the trends observed on smaller downstream datasets: LANCE retains substantial memory savings while maintaining accuracy within ~2% of full BP. These results confirm that LANCE scales effectively to ImageNet-level data.
>
> | Method | # Layers | Acc (ResNet18) | MB (ResNet18) | TFLOPs (ResNet18) | Acc (ResNet34) | MB (ResNet34) | TFLOPs (ResNet34) |
> |--------|----------|----------------|----------------|---------------------|----------------|----------------|---------------------|
> | **BP**    | 2 | 60.70 | 24.5  | 0.059 | 60.77 | 24.5  | 0.059 |
> | **BP**    | 4 | 64.28 | 61.25 | 0.090 | 64.21 | 49.0  | 0.118 |
> | **LANCE** | 2 | 59.18 | 7.12  | 0.048 | 58.59 | 4.49  | 0.046 |
> | **LANCE** | 4 | 62.20 | 16.59 | 0.075 | 62.66 | 10.79 | 0.093 |
>
>
> ## **[W3] Ablation on compression ratio vs gradient fidelity**
> We have completed the requested ablation analyzing the effect of different compression ratios on gradient fidelity. The new results (Fig. 4 of the main section and Fig. 5 of the Appendix D in the revised manuscript) show that gradient alignment degrades smoothly as the compression becomes more aggressive but consistently remains within $\sim 70^\circ$ of full BP—even under extreme compression—confirming that LANCE maintains stable descent directions across a wide range of memory budgets.
>
> ## **[Q1] Robustness when calibration data does not match target data**
> Our ablation in Figure 3 indirectly addresses this question. We show that using as few as _two_ calibration batches (≈256 images) with five different seeds yield high accuracy and low memory usage, and increasing the number of calibration batches does not significantly change performance. Since two batches represent only a tiny fraction of the CIFAR-100 training set, showing that the technique is robust.
>
> ## **[W1, Q2] On-device evaluation and small batches**
> We have now conducted small-batch fine-tuning experiments using BS=4 and BS=8 on all four datasets and both architectures. These results, included in the revised manuscript Appendix E and Tables 8 and 9, show the same accuracy, memory, and FLOPs trends as BS=128, confirming that LANCE remains effective in the tiny-batch regime typical of MCU-class devices.
>
> We are currently running real on-device experiments on a Raspberry Pi to measure actual activation memory usage and runtime. We expect to include these results before the rebuttal deadline.
>
> [1] Nguyen, L.T., Quélennec, A., Nguyen, V.T. and Tartaglione, E., Beyond Low-rank Decomposition: A Shortcut Approach for Efficient On-Device Learning. In _Forty-second International Conference on Machine Learning_.

---

> > ### Author Response · Authors · 2025-11-26
> > **Update: Real On-Device Results Completed**
> >
> > Thank you for your patience. We have now completed the real on-device experiments referenced in our earlier response.
> >
> > We implemented all methods on a Raspberry Pi 3 Model B+ (Broadcom BCM2837B0, quad-core ARM Cortex-A53) and fine-tuned an MCUNet model for 5 iterations on CIFAR-10 using batch size 128. Each experiment was repeated 5 times, and we report the mean end-to-end fine-tuning time. The updated manuscript now includes these results in **Appendix G** and **Fig. 6**.
> >
> > The table below summarizes the relative improvement of LANCE over the baselines using the total fine-tuning time (higher is better, computed as _Other / LANCE_):
> >
> > | Compared To | Min Ratio | Max Ratio | Interpretation                       |
> > | ----------- | --------- | --------- | ------------------------------------ |
> > | **BP**      | 1.03      | 1.09      | BP is 3–9% slower than LANCE         |
> > | **ASI**     | 1.19      | 1.76      | ASI is 19–76% slower than LANCE      |
> > | **HOSVD**   | 2.14      | 4.42      | HOSVD is 2.1×–4.4× slower than LANCE |
> >
> > These results confirm that LANCE's one-shot subspace calibration eliminates the runtime overhead of iterative decompositions and provides practical speedups on real edge hardware, in addition to its SRAM and FLOPs benefits.
> >
> > Please let us know if any additional details would be helpful.

---

### Author Response · Authors · 2025-11-23

We thank all reviewers for their thoughtful feedback and constructive comments. We address each point in detail below. In addition to the experiments already included in the main paper, we have now completed several of the major follow-up evaluations requested across the reviews, including:

- **Split-ImageNet fine-tuning (Table 1)**,
- **small-batch fine-tuning with BS = 4 and BS = 8 (Appendix E, Table 8 and 9)**, and
- **additional ablations analyzing compression ratio vs. gradient fidelity (Main section Figure 4, and Appendix D Figure 5)**.

All new results are incorporated in the revised manuscript and are **highlighted for visibility**. They consistently confirm the original findings. We are also conducting real on-device measurements on a Raspberry Pi to report activation memory and runtime, and we expect to include these results before the rebuttal deadline.

---

### Author Response · Authors · 2025-12-01
**Consolidated Summary of Revisions**

We thank the reviewers for their thorough engagement, which has led to significant improvements in our manuscript. To support the newly assigned Area Chair, we provide a brief summary of the revisions made in response to reviewer comments. Reviewers requested additional evaluation at larger scale, performance under small batch sizes, real on-device measurements, ablations relating compression to gradient fidelity, clarification of memory usage, and inclusion of additional continual-learning baselines.

The revised manuscript includes several major new experimental results that directly address these concerns:

* **Scalability:** Completed Split-ImageNet fine-tuning with ResNet18/34 (Table 1).
* **Small-Batch Robustness:** Conducted small-batch fine-tuning with batch sizes 4 and 8 across all datasets and architectures (Tables 8–9, Appendix E).
* **Practical Efficiency:** Added real on-device fine-tuning runtime measurements on a Raspberry Pi 3 Model B+ (Appendix G, Fig. 6).

Additional ablations and conceptual clarifications have also been incorporated:

* **Ablations:** Added analysis examining compression ratio versus gradient alignment (Fig. 4, Appendix Fig. 5) and calibration-size sensitivity (Fig. 3).
* **Memory/Robustness:** Clarified LANCE’s activation-memory behavior relative to GPM and ASI, and added discussion of subspace stability and domain shift.
* **Limitations:** A dedicated Limitations and Future Directions section has been added (Appendix F).
* **Baselines:** The continual-learning methods LMSP (UAI 2025) and SD (CVPR 2023) have been added to Table 3 and integrated into the related-work discussion.

All additions and clarifications have been incorporated into the revised manuscript and highlighted for visibility.

---

### Meta-Review · Area_Chair_NWvp · 2026-01-11

**Summary:**

This paper studies the problem of fine-tuning and on-device continual learning under memory constraints. The authors proposed LANCE (Low-rank Activation Compression) to perform one-shot higher-order SVD to obtain a reusable low-rank subspace, which reduces memory and computation by eliminating repeated decompositions. The fixed subspaces also support on-device continual learning. The proposed framework is evaluated on both standard datasets and continual learning benchmarks.

This paper receives diverse ratings. Two reviewers view this work as an innovative continual learning mechanism with a good impact for sustainable AI. In contrast, one reviewer raises concerns that the evaluation lacks real on-device results and evaluations on large datasets. Another reviewer raises concerns about the novelty of utilizing gradient projection memory, and also questions whether the Mem metric is a reasonable metric. The rebuttal is detailed and technically thorough, addressing most concrete experimental concerns, but it does not fully change the opinions of the more skeptical reviewers.

**Reviewer Concerns:**

Reviewer kGZ1 raised concerns about the lack of real on-device evaluations. The authors rebutted by adding additional on-device experiments on Raspberry. After reading the paper myself, I partially agree with the concerns. In particular, since the core motivation of this paper is reducing both memory and computation for on-device learning, more experiments on real on-device learning scenarios are quite necessary. However, the additional on-device experiments provided during rebuttal seem to lack comprehensiveness and are inadequate to fully support the central claim.

Reviewer u2Zf raised concerns that Phase 3 is similar to GPM and thus weakening the overall novelty. Authors rebutted that LANCE is a three-stage framework and no prior work has explored low-rank orthogonal subspaces. However, a direct distinction between Phase 3 and GPM is not clearly illustrated in the rebuttal, and the authors also acknowledge that Phase 3 may resemble GPM when viewed in isolation. Therefore, this concern is not adequately addressed.

**Reviewer Scores:**

Reviewer kGZ1: Assigned a clear rejection and raised concerns about the lack of on-device experiments and large-scale evaluations. While authors provide additional on-device experiments, these results seem to be insufficient to provide thorough support. Therefore, although no explicit post-rebuttal update was provided, a score change toward positive ratings is unlikely.

Reviewer TyNZ: Rated the paper marginally above the acceptance threshold and retained the original score during discussion, saying that the issue of robustness to domain shift is partially addressed. Based on the comments, the score would likely remain unchanged.

Reviewer tjoS: Rated the paper marginally above the acceptance threshold and retained the original score during the discussion with authors. Therefore, a score change is unlikely.

Reviewer u2Zf: Rated the paper marginally below the acceptance threshold and did not participate in the discussion. While the authors emphasized that LANCE is a three-stage framework, the response did not clearly distinguish Phase 3 from GPM. Therefore, the score is likely to be unchanged.

---

### Decision · Program_Chairs · 2026-01-26

Reject